# Unraveling surface sensitivity for generating metastable active sites in molybdenum-based catalysts for CO$_2$ hydrogenation

Yifan Feng[1,4], Zhenyu Xing [1,4], Daoping Ye[2], Jin Niu[1], Yu Tian[1], Tian Ma [1], Chong Cheng [1]✉, Bo Yin[1], Arne Thomas [3]✉ & Shuang Li [1]✉

The reverse water-gas shift (RWGS) reaction is crucial for sustainable CO$_2$ conversion, yet catalyst surface remodeling at high temperatures remains a complex and pivotal phenomenon. This study investigates the complex relationship between surface reconstruction and catalytic performance using a series of molybdenum-based catalysts, which can generate different catalytic MoO$_3$ surface layers under RWGS conditions. In-situ characterization techniques and theoretical analyses reveal that the MoO$_3$ layer on MoO$_3$/MoO$_2$-C and MoO$_3$/Mo$_2$N-C is in-situ reduced to MoO$_2$ and metastable MoO$_x$ (2 < x < 3), respectively, while it is not reduced on MoO$_3$/Mo$_2$C-C during the catalysis process. The metastable MoO$_x$ species on MoO$_3$/Mo$_2$N-C shows an unprecedented CO yield (up to 48.3 %) nearing the equilibrium conversion limit, a CO formation rate of $8.26 \times 10^{-5}$ mol$_{CO}$ g$_{cat}^{-1}$ s$^{-1}$, and 99 % CO selectivity under 500 °C. The function and formation conditions of metastable MoO$_x$ sites are comprehensively investigated in this work.

Thermal catalytic reduction of CO$_2$ into value-added products has been considered one of the most promising strategies for achieving carbon-neutralization goals[1–3]. Worldwide initiatives are aimed at establishing a sustainable hydrogen infrastructure[4–6], which has positioned the hydrogenation of carbon dioxide as a viable route for CO$_2$ utilization[7–9]. However, due to the stability of CO$_2$ molecules, the transformation of CO$_2$ usually relies on highly active heterogeneous catalysts[10]. The production of carbon monoxide (CO) via the reverse water gas shift reaction (RWGS) is a reaction of immense economic and ecological importance. This is because the resulting CO or synthesis gas (a mixture of CO and H$_2$) can serve as fundamental components for synthesizing a wide array of chemicals within sequential catalytic processes[11]. Economic analyses suggest that pathways that use CO$_2$ to generate CO could offer competitive advantages over other methods, especially in scenarios where renewable energy sources are abundant[12,13]. This underscores the strategic importance of developing and optimizing RWGS reactions and their integration into the broader chemical production industry[14–17].

To fully harness the potential of the RWGS reaction on a global scale, the catalyst development needs to meet important criteria[18–20]. For instance, high CO selectivity is required to prevent hydrogen from being lost to methane or other side-products, which simplifies the downstream separation process[21–23]. However, due to the endothermic nature of the reaction, relatively high operating temperatures, of up to 800 °C, and ideally high H$_2$/CO$_2$ ratios are required to achieve a satisfactory conversion[10]. Such reaction conditions often lead to sintering and agglomeration of the catalysts, particularly with commonly used high-content Cu- or noble metal-based catalysts, ultimately resulting in poor stability during long-term practical use[24–27]. Recently, numerous efforts have been devoted to developing new catalyst systems for RWGS reactions with high activity and stability. In this regard, some progress has also been achieved for Mo-based catalysts. For example, MoP supported on CNTs has been reported for RWGS. While this catalyst has a low CO$_2$ conversion rate of only 16% it showed 100% CO selectivity at 500 °C under the weight hourly space velocity (WHSV) of 36,000 mL g$_{cat}^{-1}$ h$^{-1}$ [28]. Single-atom molybdenum supported

[1]College of Polymer Science and Engineering, State Key Laboratory of Advanced Polymer Materials, Sichuan University, Chengdu, China. [2]College of Chemical Engineering, Sichuan University, Chengdu, China. [3]Functional Materials, Department of Chemistry, Technische Universität Berlin, Berlin, Germany. [4]These authors contributed equally: Yifan Feng, Zhenyu Xing. ✉e-mail: chong.cheng@scu.edu.cn; arne.thomas@tu-berlin.de; shuang.li@scu.edu.cn

on nitrogen-doped carbon (Mo/NC) has also been investigated as RWGS catalyst, achieving a $CO_2$ conversion rate of 30.4% with nearly 100% CO selectivity at 500 °C, even under extremely low $H_2$ partial pressure[29]. Recently, nanocrystalline cubic molybdenum carbide (α-$Mo_2C$) has demonstrated remarkable performance, achieving approximately 60% $CO_2$ conversion and 100% CO selectivity at high space velocity, even after over 500 h of exposure to harsh reaction conditions at 600 °C[30]. These studies indicate that Mo-based materials are very promising catalysts for RWGS in practical applications. However, the performance of different Mo components differs a lot when their supports, structures, synthesis conditions, and reaction conditions change[2,31–33]. This poses a great challenge to the reasonable design of high-performance Mo-based catalysts and a precise understanding of its mechanism, especially under high temperatures, thereby hindering the further development of Mo-based RWGS catalysts. We have recently reported that organic-polyoxometallate precursors form molybdenum nitrides or carbides, depending on the carbonization temperature[32]. This precursor offers the possibility of investigating and solving the challenges mentioned above. Herein, we designed a series of catalytic $MoO_3$ surface layers on different molybdenum catalysts from an organic-polyoxometalate crystal precursor, prepared via condensation of ammonium molybdate and p-phenylenediamine, as a system for a comprehensive investigation of the catalytic mechanisms and the reconstruction behavior of metastable active sites under RWGS conditions. By adjusting the carbonization temperatures, the organic-polyoxometalate precursor transformed into $MoO_2$, $Mo_2N$, and $Mo_2C$ nanocrystals on a carbon matrix, named $MoO_2$-C, $Mo_2N$-C, and $Mo_2C$-C. When exposed to air during the synthesis process, a uniform surface oxidization layer of $MoO_3$ is formed on these catalysts (i.e., $MoO_3/MoO_2$-C, $MoO_3/Mo_2N$-C, and $MoO_3/Mo_2C$-C), therefore providing materials which allow the systematic investigation of the surface and bulk catalytic properties of different Mo compounds. Comprehensive in-situ characterization techniques and theoretical analyses revealed that the $MoO_3$ layer on $MoO_3/MoO_2$-C and $MoO_3/Mo_2N$-C will be in-situ reduced to $MoO_2$ and metastable $MoO_x$ ($2 < x < 3$), respectively, while it is not reduced on $MoO_3/Mo_2C$-C during the catalysis process. The activity of these catalysts for the RWGS reaction has been investigated, which indicates that the metastable $MoO_x$ ($2 < x < 3$) species on $MoO_3/Mo_2N$-C shows an unprecedented $CO_2$ conversion (up to 48.3 %) close to the equilibrium conversion limit, a CO formation rate of $8.26 \times 10^{-5}$ $mol_{CO}$ $g_{cat}^{-1}$ $s^{-1}$, and up to 99% CO selectivity at 500 °C. In this work, the formation conditions of metastable $MoO_x$ active sites and their function for catalysis in RWGS have been comprehensively investigated. The strategy of constructing highly active metal oxide surfaces by adjusting the reducibility of the substrate can provide new insights into the design of high-performance heterogeneous catalysts for various reactions, such as the RWGS and Fischer–Tropsch processes.

## Results

### Structure characterization of molybdenum-based catalysts

To verify that the as-prepared catalysts have the expected structure as shown in Fig. 1a, comprehensive characterizations were carried out. Scanning electron microscopy (SEM) measurements were first conducted to reveal the microscale morphologies of the as-synthesized catalysts. As shown in Supplementary Fig. 1, the uniform nanosheet structures from the metal-organic precursors have been maintained in the pyrolyzed samples. The surface oxidation structure was probed using Raman spectroscopy, showing characteristic $MoO_3$ peaks present across all three catalysts. Notably, $MoO_3/Mo_2N$-C exhibits the most intense peaks, indicative of a higher $MoO_3$ content on the $Mo_2N$ surface (Fig. 1b–d). Powder X-ray diffraction (XRD) was used to describe the crystallographic characteristics of the catalysts. The $MoO_3/Mo_2N$-C and $MoO_3/Mo_2C$-C exhibit distinct diffraction peaks aligning with the known patterns of $Mo_2N$ (PDF no. 25-1366) and $Mo_2C$

(PDF no. 44-1159) (Fig. 1b–d). Conversely, the $MoO_3/MoO_2$-C displays just a broad peak around 25°, suggesting the mainly amorphous nature of the $MoO_2$ component with only small crystalline areas (vide infra).

Further investigation using high-angle annular dark-field scanning transmission electron microscopy (HAADF-STEM) illuminates the consistent nano-sized crystal arrangement interconnected by the carbon substrate, forming a coherent 2D nanosheet structure. Crystalline structures are discerned with clear lattice fringes corresponding to the planes of $Mo_2N$ (220), $MoO_2$ (−211), and $Mo_2C$ (100), with respective spacings of 0.154, 0.256, and 0.253 nm, which confirms the precise control over the bulk phase of the catalysts (Fig. 1e–g i, Supplementary Figs. 2–4). Based on the corresponding HAADF images with false color and the surface plot (Fig. 1e–g ii, v), the thickness distribution on the surfaces of the three catalysts can be obtained, which allows for a preliminary assessment of the distribution of $Mo_2N$, $MoO_2$, $Mo_2C$, and $MoO_3$ on the three catalysts. To obtain more accurate distribution information, geometric phase analysis (GPA) was further conducted on the $MoO_3/Mo_2N$-C, $MoO_3/MoO_2$-C, and $MoO_3/Mo_2C$-C catalysts (Fig. 1e–g iii). The strain tensor component $\varepsilon_{xx}$ reveals that lattice strain exists in all three catalysts, which is a result of the interactions between $Mo_2N$, $MoO_2$, $Mo_2C$, and $MoO_3$ (Red represents tensile stress, while blue indicates compressive stress). Due to these interactions, the intrinsic stress of $Mo_2N$, $MoO_2$, and $Mo_2C$ differs from that of $MoO_3$. The distribution of molybdenum species can be more accurately determined by combining the results from Fig. 1e–g i-iii as shown in the overlay in Fig. 1e–g iv and the 3D surface plots shown in Fig. 1e–g v. Elemental mapping via energy-dispersive X-ray spectroscopy (EDS) from HAADF-STEM provide evidence of the uniform distribution of Mo, N, O, and C across the materials, as visualized in Supplementary Figs. 5–7.

X-ray photoelectron spectroscopy (XPS) was conducted to probe the electronic configuration and oxidation states of the $MoO_3/Mo_2N$-C, $MoO_3/MoO_2$-C, and $MoO_3/Mo_2C$-C catalysts (Supplementary Figs. 8, 9). Figure 2a presents the high-resolution Mo 3$d$ spectra of these catalysts, where $MoO_3/Mo_2N$-C and $MoO_3/Mo_2C$-C show apparent peaks of $Mo^{\delta+}$ at 228.8 and 231.9 eV, indicating the existence of $Mo_2N$ and $Mo_2C$ as main components. In contrast, in $MoO_3/MoO_2$-C, only the peaks for $Mo^{4+}$ ($MoO_2$, 229.4 and 233.2 eV) and $Mo^{6+}$ ($MoO_3$, 232.7 and 235.8 eV) can be observed, indicating that the stoichiometric Mo oxides are the main component for $MoO_3/MoO_2$-C. All three catalysts exhibit distinct $Mo^{6+}$ peaks, confirming the presence of surface $MoO_3$ on each material. This result aligns consistently with the Raman characterization data.

The reducibility of the $MoO_3$ surface layer on the different substrates has been assessed by temperature-programmed reduction in $H_2$ ($H_2$-TPR). A pronounced $H_2$ uptake peak for $MoO_3/Mo_2N$-C catalyst at 317 °C is noticeable, significantly lower than the peaks at 342 °C for $MoO_3/MoO_2$-C and 431 °C for $MoO_3/Mo_2C$-C, as illustrated in Fig. 2b. The more intense and lower-temperature reduction peak for $MoO_3/Mo_2N$-C implies easier reducibility of surface $MoO_3$, conducive to form an active surface layer. Additionally, a secondary peak at 372 °C for $MoO_3/MoO_2$-C suggests a deeper reduction step. This may involve the reduction of $MoO_3$ on the $MoO_3/MoO_2$-C surface to $MoO_x$ under an $H_2$ atmosphere, followed by its subsequent reduction to $MoO_2$[34].

We furthermore systematically examined the surface structural evolution under reductive conditions through in situ Raman spectroscopy. As shown in Fig. 2c, after the $H_2$ treatment of the $MoO_3/Mo_2N$-C catalyst at room temperature, the surface $MoO_3$ is converted to $MoO_x$. It can be assumed that under an $H_2$ atmosphere, some lattice oxygen is released from the $MoO_3$ on the surface of $MoO_3/Mo_2N$-C, forming $MoO_x$. Due to the interactions between $MoO_3$ and $Mo_2N$, $MoO_3$ can be reduced by $H_2$, but not sufficient to be fully reduced to $MoO_2$, resulting in a phase where $2 < x < 3$[35]. Changing the atmosphere again to air at room temperature lead to the return of

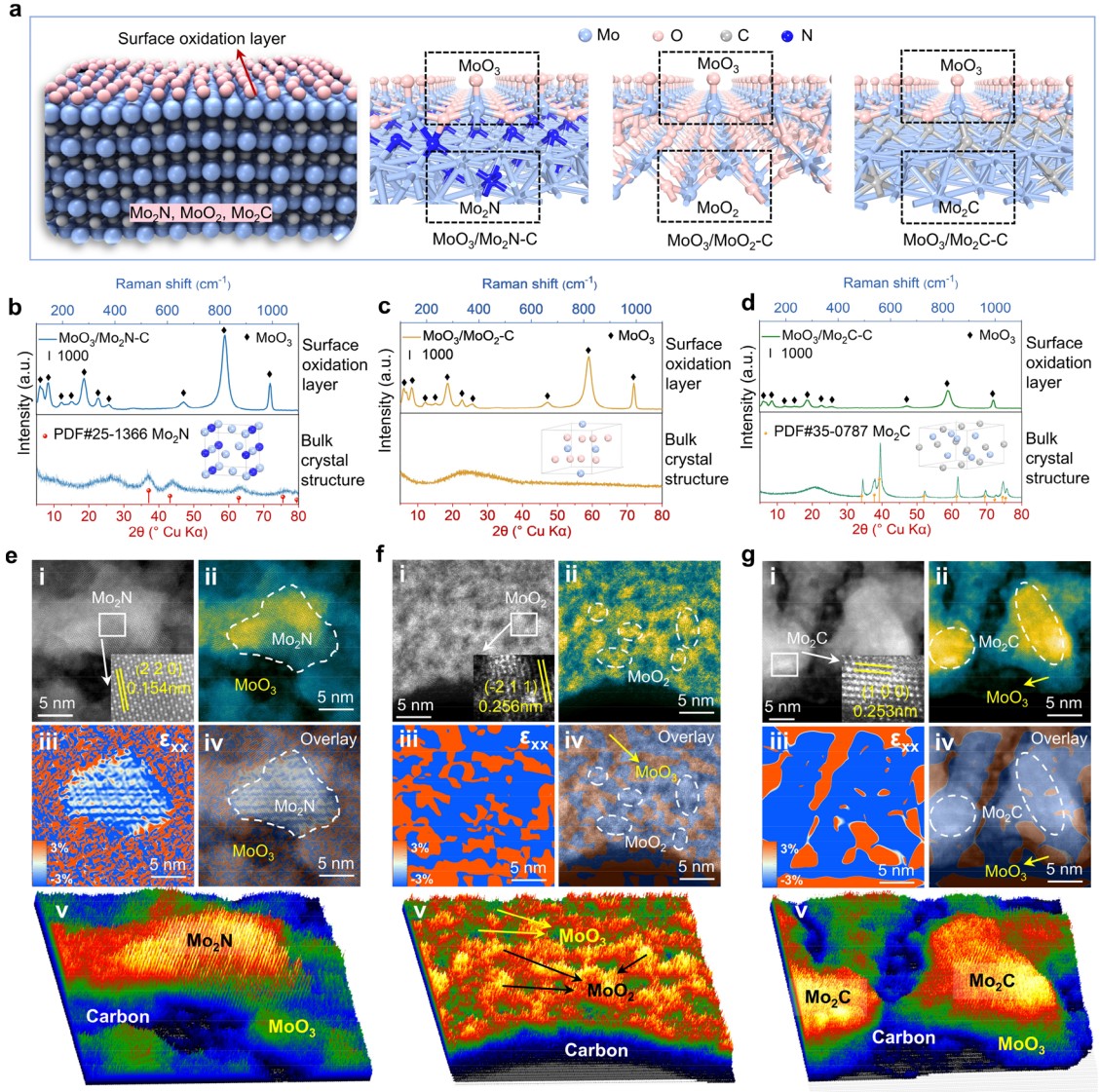

**Fig. 1 | Characterization of the heterostructured molybdenum-based catalysts.** **a** Schematic illustration of the catalyst's surface structure. Light blue: molybdenum atoms; dark blue: nitrogen atoms; pink: oxygen atoms; gray: carbon atoms. **b**–**d** XRD patterns and Raman spectra. The inset shows the model schematic diagram of $Mo_2N$, $MoO_2$, and $Mo_2C$; **e**–**g** HAADF-STEM images and elucidation of the heterojunctions in $MoO_3/Mo_2N$-C, $MoO_3/MoO_2$-C, and $MoO_3/Mo_2C$-C. i) HAADF-STEM images; ii) false color image of the HAADF-STEM image from i); iii, iv) lattice strain map of $MoO_3/Mo_2N$-C, $MoO_3/MoO_2$-C, and $MoO_3/Mo_2C$-C from i); v) 3D surface plot of $MoO_3/Mo_2N$-C, $MoO_3/MoO_2$-C, and $MoO_3/Mo_2C$-C from i).

the surface $MoO_x$ phase back to $MoO_3$, highlighting the pronounced sensitivity of the surface $MoO_3$ on the $Mo_2N$ substrate to ambient $O_2$ or reductive $H_2$ conditions, as corroborated by Supplementary Fig. 10, showing the Raman spectra when the atmosphere is changed from $H_2$ to air. Conversely, on the $MoO_3/MoO_2$-C catalysts, in-situ $H_2$ treatment at room temperature exclusively yields $MoO_2$ peaks, signifying that the metastable $MoO_x$ species could not be stabilized, and the $MoO_3$ is further reduced to $MoO_2$ on the $MoO_2$ substrate, as depicted in Fig. 2d. Note that in comparison to $MoO_3/Mo_2N$-C, only for $MoO_3/Mo_2C$-C the characteristic peaks of $MoO_3$ can be observed in the in situ Raman spectra(Fig. 2e).

Furthermore, density functional theory (DFT) calculations were performed to gain deeper insights into the underlying mechanisms of the surface reconstruction phenomenon observed in the in-situ experiments. As shown in Supplementary Fig. 11, the obvious orbital overlap in the partial density of states (pDOS) plots reveals a strong interaction between the substrate ($MoO_2$, $Mo_2N$, $Mo_2C$) and surface oxide layer ($MoO_3$) in all cases. The calculated formation energy of an

oxygen vacancy in Fig. 3a indicates that $MoO_3/Mo_2N$-C own a much lower energy barrier (0.37 eV) to lose oxygen from the oxide layer when compared with $MoO_3/MoO_2$-C (1.25 eV) and $MoO_3/Mo_2C$-C (1.95 eV), which is consistent with the sequence observed in the $H_2$-TPR results, indicating that $Mo_2N$ more readily forms oxygen vacancies in supported $MoO_3$. When the substrate ($Mo_2N$, $MoO_2$, $Mo_2C$) comes into contact with the surface layer ($MoO_3$), different electronic interactions arise, thereby influencing the properties of the surface layer. The calculation results show that in $MoO_3/Mo_2N$-C, the Mo-O bonds in the surface $MoO_3$ layer are significantly elongated with a value of 1.98 Å resulting from the strong interaction between $Mo_2N$ and $MoO_3$, weakening the strength of the Mo-O bonds; while $MoO_3/MoO_2$-C and $MoO_3/Mo_2C$-C exhibit a relatively shorter Mo-O bond length of 1.89 Å and 1.80 Å, respectively (insets in Fig. 3a). Moreover, the pDOS plots in Fig. 3b indicate that the $MoO_3$ layer in $MoO_3/Mo_2N$-C exhibits a relatively prominent electronic state near the Fermi level, which facilitates electron transfer during the reaction and promotes the formation of oxygen vacancies, as further supported by the oxygen vacancy

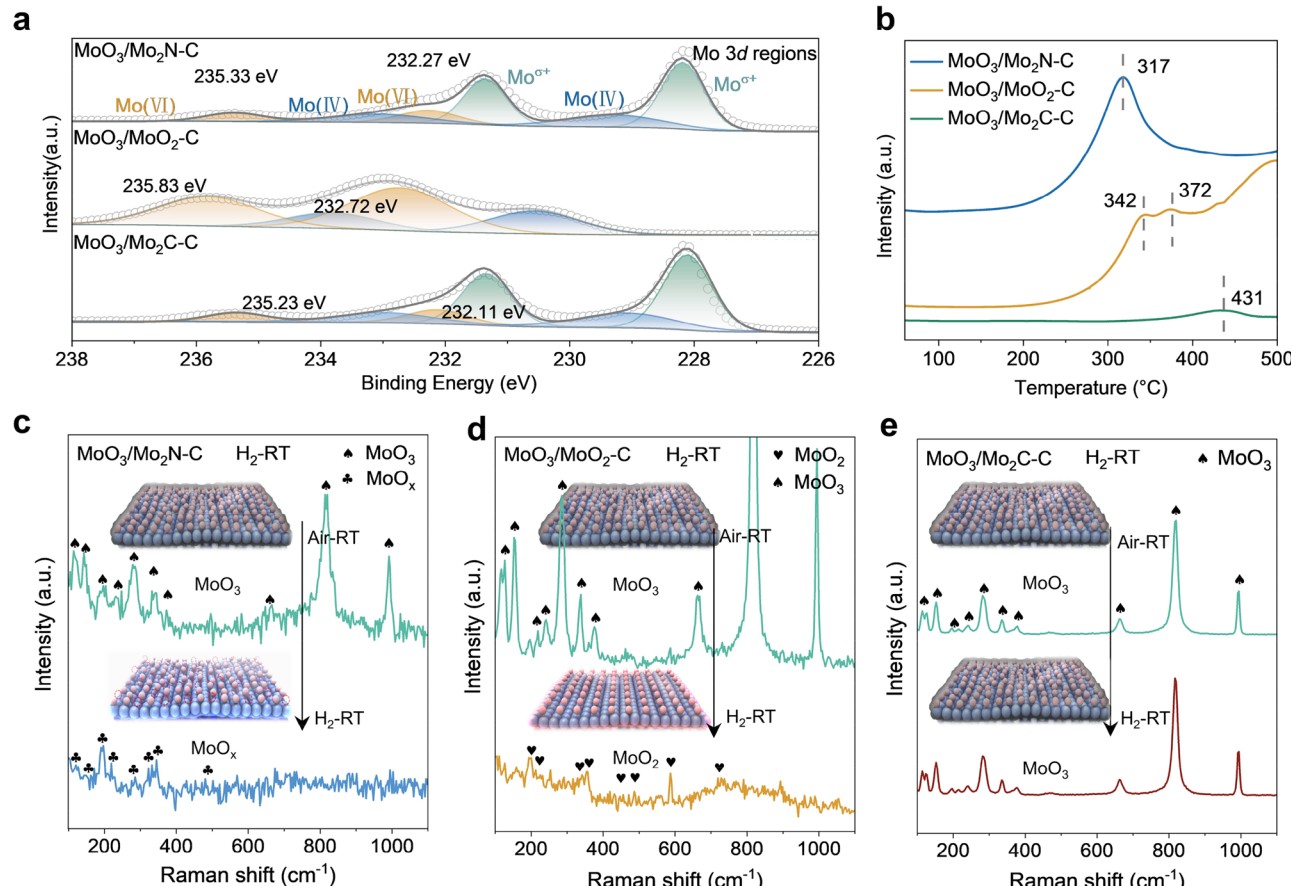

**Fig. 2 | Electronic structure analysis and surface reconstruction mechanism investigation of the MoO₃/Mo₂N-C, MoO₃/MoO₂-C, and MoO₃/Mo₂C-C catalysts. a** Curve-fitted high-resolution XPS Mo 3*d* spectra of the different materials. **b** H₂-TPR of MoO₃/Mo₂N-C, MoO₃/MoO₂-C, and MoO₃/Mo₂C-C catalysts. **c–e** In-situ Raman results of the materials under H₂ pretreatment. The inset shows schemes of the surface structure of three catalysts in different atmospheres.

formation energy calculations shown in Fig. 3a. However, in MoO₃/MoO₂-C and MoO₃/Mo₂C-C, the strong Mo−O bonds and limited surface electron transfer, reflected by their larger band gaps, make the formation of MoOₓ species more difficult. Surface reconstruction under flowing hydrogen conditions is shown in Fig. 3c.

**Reactant activation on the catalysts**

Molecular dynamics (MD) and DFT simulations were performed to investigate the CO₂ adsorption properties on MoO₃/Mo₂N-C, MoO₃/MoO₂-C, and MoO₃/Mo₂C-C catalysts under RWGS reaction conditions (Supplementary Fig. 12). The distances distribution of CO₂ molecules on the catalyst surface is statistically analyzed within 1.5 nm, where MoO₃/Mo₂N-C exhibits the highest proportion of gas molecules in close distance to the surface (around 4 Å) indicating the strongest CO₂ adsorption capability on the MoO₃/Mo₂N-C surface (Supplementary Fig. 13). As shown in Fig. 4a–c and Supplementary Fig. 14a–c, the calculated adsorption energy demonstrates the strong binding ability of CO₂ on MoO₃/Mo₂N-C (−0.42 eV) compared to MoO₃/MoO₂-C (−0.26 eV) and MoO₃/Mo₂C-C (−0.23 eV). Moreover, theoretical simulations indicate that the CO₂ molecule exhibits obvious activation upon adsorption on the MoO₃/Mo₂N-C surface, which is reflected in the bending of molecular configuration, the elongation of chemical bonds, and the gain of electron density from Mo sites. The differential electron density analysis in Fig. 4a–c and Supplementary Fig. 14 show that the CO₂ molecule adsorbed on MoO₃/Mo₂N-C owns the largest angular bending (from 180.00° to 140.75°), the longest C=O bond length (from 1.18 Å to 1.28 Å), and the maximum electron transfer (0.63 |e|), while the adsorption of CO₂ is much weaker on both MoO₃/MoO₂-C and MoO₃/

Mo₂C-C, on which the adsorbed CO₂ molecule exhibits a linear configuration. PDOS plots (Fig. 4d–f and Supplementary Fig. 15) reveal that the electron state distribution changes significantly, with a substantial overlap of states between Mo and CO₂ after the adsorption of CO₂ on MoO₃/Mo₂N-C, further supporting the above results. Similarly, the adsorption and activation of H₂ on MoO₃/Mo₂N-C exhibit the same results when compared with MoO₃/MoO₂-C and MoO₃/Mo₂C-C (Supplementary Figs. 16, 17). These calculations show that the unique Mo₂N substrate structure endows the surface MoOₓ center with enhanced CO₂ and H₂ activation capabilities, facilitating subsequent catalytic reactions. Figure 4g schematically summarizes the differences in CO₂ activation for the three catalysts as a result of different surface reconstructions.

**CO₂ hydrogenation performances and surface reconstructions in the RWGS reaction**

We assessed the CO₂ hydrogenation catalytic efficiency of the MoO₃/Mo₂N-C, MoO₃/MoO₂-C, and MoO₃/Mo₂C-C catalysts in the RWGS reaction, both before and after H₂ pretreatment, within a fixed-bed reactor. The tests were conducted at a WHSV of 60,000 mL·g_{cat}⁻¹·h⁻¹. The MoO₃/Mo₂N-C catalyst shows an initial CO yield of 10.4 % at 400 °C, which increases significantly to 31.3% after H₂ pretreatment, as depicted in Fig. 5a. At an elevated reaction temperature of 500 °C, the yield further rises to 48.3% and thus approaches the equilibrium conversion limit under the given WHSV. In contrast, the H₂-pretreated MoO₃/MoO₂-C and MoO₃/Mo₂C-C catalysts yield considerably lower CO efficiencies of 15.8% and 4.9% at 400 °C and after H₂ pretreatment, respectively (Fig. 5b, c). At 500 °C under the same WHSV, a CO₂ conversion of 48.2%, 36.5%, 12.6% and CO formation rate of 8.26 × 10⁻⁵,

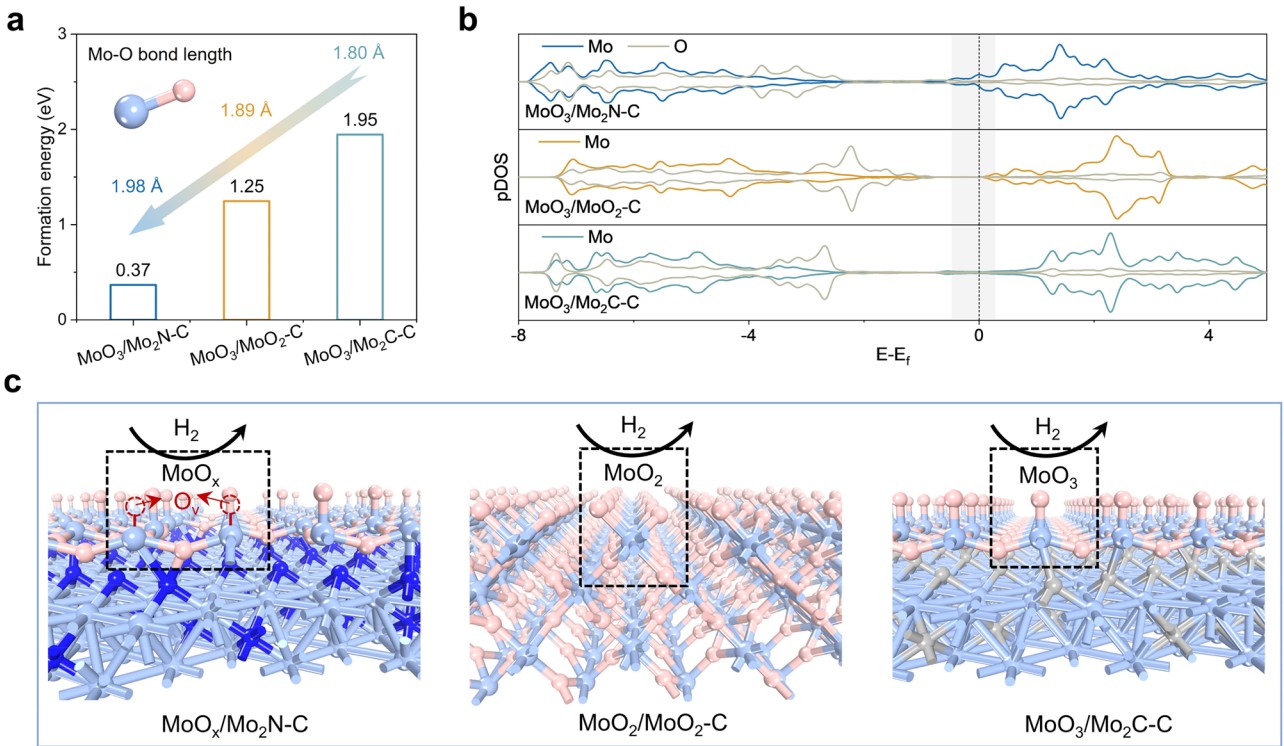

**Fig. 3 | DFT calculations reveal the intrinsic mechanism of oxygen vacancy formation. a** Calculated formation energy of oxygen vacancy and Mo-O ($MoO_3$) bond length of different materials. **b** PDOS analysis of the Mo-O ($MoO_3$) bond for different materials. **c** The structural schematic diagrams after surface reconstruction under flowing $H_2$.

$6.16 \times 10^{-5}$, $2.11 \times 10^{-5}$ $mol_{CO2}$ $g_{cat}^{-1}$ $s^{-1}$ is obtained for $MoO_3/Mo_2N$-C, $MoO_3/MoO_2$-C, $MoO_3/Mo_2C$-C catalysts, respectively (Fig. 5d, e). Notably, all catalysts exhibit exceptional CO selectivity exceeding 99 % across varying temperatures and space velocities. (Fig. 5f). In addition, the structure of the supports ($Mo_2N$, $MoO_2$, $Mo_2C$) of the three catalysts do not change during the $H_2$ treatment process (Supplementary Fig. 18). The apparent activation energy (Ea) of the $MoO_3/Mo_2N$-C catalyst is determined to be around 36 kJ mol$^{-1}$, which is significantly lower than that of $MoO_3/MoO_2$-C (47 kJ mol$^{-1}$) and only two-thirds of that of $MoO_3/Mo_2C$-C (58 kJ mol$^{-1}$). The resulting higher catalytic efficiency of $MoO_3/Mo_2N$-C is thus closely related to surface reconfiguration of $MoO_3$ to $MoO_x$ during the reaction (Fig. 5g). As shown in Fig. 5h, i, the reaction orders for $H_2$ are 0.42, 0.62 and 0.64 and for $CO_2$ 0.36, 0.45, and 0.47 for $MoO_3/Mo_2N$-C, $MoO_3/MoO_2$-C, $MoO_3/Mo_2C$-C, respectively. The higher reaction orders of $MoO_3/MoO_2$-C and $MoO_3/Mo_2C$-C for $CO_2$ and $H_2$ than for $MoO_3/Mo_2N$-C suggest that the adsorption and activation of reactant molecules are more hindered on their surfaces than for $MoO_3/Mo_2N$-C. On the other hand, the low $CO_2$ reaction order on $MoO_3/Mo_2N$-C shows that the presence of $MoO_x$ promotes the adsorption and activation of reactant molecules. We conducted stability tests on the best-performing catalysts. As shown in Supplementary Fig. 19, the $MoO_3/Mo_2N$-C catalyst exhibits excellent stability, maintaining more than 96.7% of its initial activity after the 200-hour evaluation.

To investigate the impact of surface molybdenum species reconstruction on activity in our catalysts, we controlled variables including particle size, specific surface area, and carbon content across the three catalysts. Since all catalysts were obtained from the same precursor via different calcination temperatures, their physical properties (e.g., particle size, surface area) are highly similar. As shown in Supplementary Figs. 20, 21, particle size distribution histograms and Brunauer-Emmett-Teller (BET) surface areas calculated from nitrogen adsorption-desorption isotherms confirm that $MoO_3/Mo_2N$-C, $MoO_3/MoO_2$-C, and $MoO_3/Mo_2C$-C exhibit comparable particle sizes

($1.62 \pm 0.49$ nm, $1.68 \pm 0.61$ nm, and $1.99 \pm 1.30$ nm) and specific surface areas ($2.60$ m$^2 \cdot$g$^{-1}$, $2.38$ m$^2 \cdot$g$^{-1}$, and $4.08$ m$^2 \cdot$g$^{-1}$). Furthermore, XPS analysis (Supplementary Fig. 22) quantifies the carbon layer proportion in each catalyst. Due to carbon contributions from both the carbon layer and molybdenum carbide in $MoO_3/Mo_2C$-C, we included pure $Mo_2C$ XPS as a reference, estimating the carbon content within the carbon layer of $MoO_3/Mo_2C$-C. As shown in Supplementary Fig. 22b, the carbon content in pure $Mo_2C$ is ~44.4 at. %. Therefore, the carbon content in the carbon layer of the $MoO_3/Mo_2C$-C catalyst can be estimated at ~31.6 at.%, indicating that the carbon layer content is similar across all three catalysts.

To compare and understand the specific role of the oxide layer on the catalyst surface on $CO_2$ hydrogenation, we synthesized $Mo_2N$-C (500 °C, Ar, 2 h) and $MoO_2$-C (700 °C, Ar, 2 h) in situ under identical reactor conditions, preventing surface $MoO_3$ formation by avoiding air exposure, and immediately conducted RWGS reactions. Thermogravimetric analysis (TGA) determines precise feedstock loading to ensure consistent reaction conditions between bare supports and oxide-coated catalysts (Supplementary Fig. 23). Specifically, when the WHSV is 60,000 mL·gcat$^{-1}$h$^{-1}$ and the catalyst feed mass is 100 mg, the precursor mass loss at 500 °C is 79.7%. When preparing the $Mo_2N$-C catalyst, the precursor feed mass is 125 mg. At 700 °C, the mass loss of the precursor is 56.1%, and the precursor feed mass is 178 mg when preparing the $MoO_2$-C catalyst. Furthermore, XRD characterization of synthesized $Mo_2N$-C and $MoO_2$-C (Supplementary Fig. 24) shows unchanged diffraction peaks, confirming preserved lattice structures, indicating that $Mo_2N$-C and $MoO_2$-C differ from $MoO_3/Mo_2N$-C and $MoO_3/MoO_2$-C solely by lacking the oxide layer.

Supplementary Fig. 25 shows that the $CO_2$ conversion rate and CO yield of the bare support $Mo_2N$-C and $MoO_2$-C without a surface $MoO_3$ oxide layer are both lower than those of $MoO_3/Mo_2N$-C and $MoO_3/MoO_2$-C. After eliminating minor differences in feedstock quantity through the CO formation rate, it can be observed that the CO

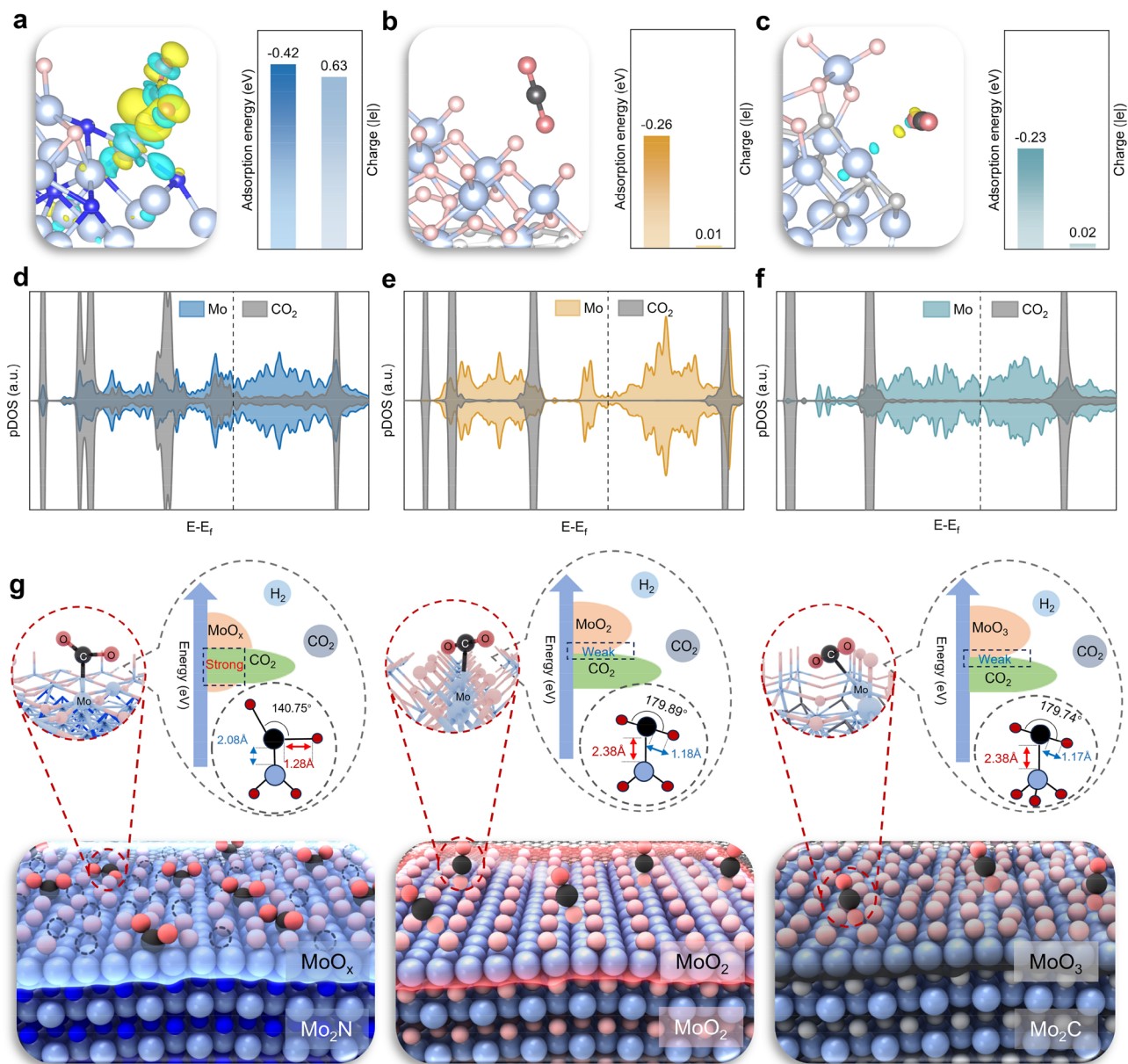

**Fig. 4 | Surface reactant activation on the different heterostructures and metastable $MoO_x$ site.** Differential electron density plot, calculated adsorption energy and Bader charge of $CO_2^*$ on **a** $MoO_3/Mo_2N$-C, **b** $MoO_3/MoO_2$-C, and **c** $MoO_3/Mo_2C$-C (cyan and yellow show charge consumption and accumulation, respectively, the cut-off of the density-difference iso-surface is 0.004 e·Bohr$^{-3}$;

color codes: light blue, molybdenum; blue, nitrogen; pink, oxygen in lattice; red, oxygen in molecule; gray, carbon in lattice; darkgray, carbon in molecule). PDOS analysis of $CO_2^*$ on **d** $MoO_3/Mo_2N$-C, **e** $MoO_3/MoO_2$-C, and **f** $MoO_3/Mo_2C$-C. **g** Schematic illustration of the $CO_2^*$ on the catalyst surfaces.

formation rate of $Mo_2N$-C is reduced from $8.26 \times 10^{-5}$ $mol_{CO2}$ $g_{cat}^{-1}$ $s^{-1}$ to $5.19 \times 10^{-5}$ $mol_{CO2}$ $g_{cat}^{-1}$ $s^{-1}$ compared to $MoO_3/Mo_2N$-C, and the CO formation rate for $MoO_2$-C is reduced from $6.17 \times 10^{-5}$ $mol_{CO2}$ $g_{cat}^{-1}$ $s^{-1}$ to $4.80 \times 10^{-5}$ $mol_{CO2}$ $g_{cat}^{-1}$ $s^{-1}$ compared to $MoO_3/MoO_2$-C. The performance of the bare support $Mo_2N$-C decreases more significantly. Additionally, as shown in Supplementary Fig. 26, we have included measurements of activation energies and reaction orders for oxide-free $Mo_2N$-C and $MoO_2$-C. The apparent activation energy ($E_a$) of the $Mo_2N$-C catalyst is determined as 47.96 kJ·mol$^{-1}$, significantly lower than that of $MoO_2$-C (65.00 kJ·mol$^{-1}$). Concurrently, the $H_2$ reaction order for $Mo_2N$-C (0.36) is markedly lower than for $MoO_2$-C (0.61), following the same trend observed for the $MoO_3/Mo_2N$-C and $MoO_3/MoO_2$-C catalysts. However, $Mo_2N$-C and $MoO_2$-C exhibit similar $CO_2$ reaction orders (0.64 and 0.71), revealing a distinct pattern. This

indicates that the surface $MoO_3$ layer has minimal impact on the $H_2$ reaction order but significantly modulates the $CO_2$ reaction order.

To clarify the role of the oxide layer, we compared $CO_2$ and $H_2$ reaction orders and activation energies between catalysts with and without surface oxides. As shown in Supplementary Fig. 27, the $CO_2$ reaction order for $MoO_3/Mo_2N$-C is 0.36, while the $Mo_2N$-C exhibits an increased $CO_2$ reaction order of 0.64; similarly, $MoO_3/MoO_2$-C shows a $CO_2$ reaction order of 0.45 versus $MoO_2$-C at 0.71. The lower $CO_2$ reaction orders signify superior $CO_2$ activation capability in $MoO_3/Mo_2N$-C and $MoO_3/MoO_2$-C catalysts, which originates directly from the surface $MoO_3$ layer. Concurrently, the negligible difference in $H_2$ reaction orders confirms that $H_2$ activation capability is independent of the oxide layer. Notably, oxide-free catalysts exhibit higher activation energies (Supplementary Fig. 28), likely due to impaired $CO_2$ adsorption and activation without surface $MoO_3$, resulting in higher temperature dependency.

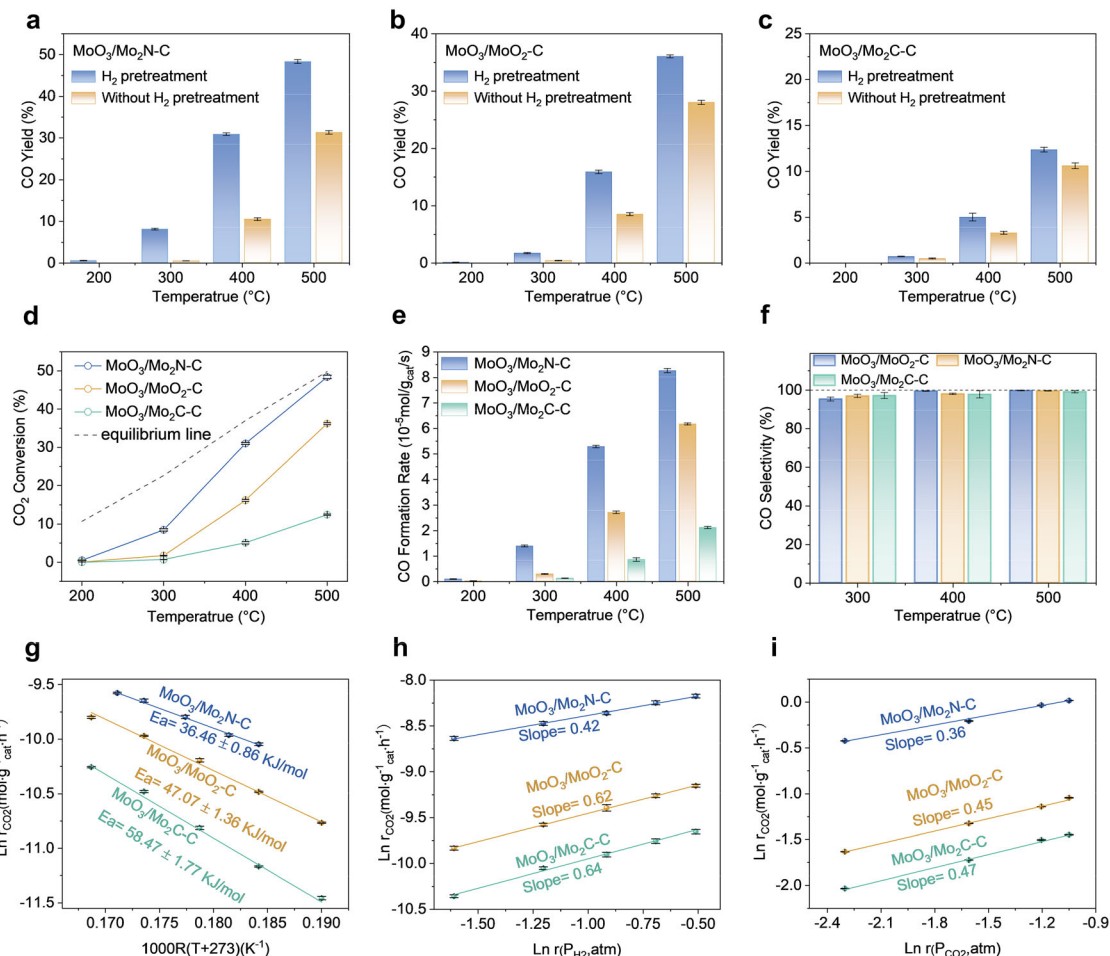

**Fig. 5 | CO$_2$ hydrogenation performances of molybdenum-based catalysts.**
**a**–**c** Catalytic results for CO yield under different temperatures, **d** CO$_2$ conversion at various temperatures under 60,000 mL·g$_{cat}^{-1}$h$^{-1}$, **e** Catalytic reaction rates, **f** CO selectivity, **g** Apparent activation energy (Ea), **h**, **i** Kinetic orders of reactants (H$_2$ and CO$_2$) for MoO$_3$/Mo$_2$N-C, MoO$_3$/MoO$_2$-C, and MoO$_3$/Mo$_2$C-C catalysts. ($n$ = 3 independent experiments, data are presented as mean values ± SD).

To elucidate the catalytic nature of MoO$_3$/Mo$_2$N-C, we comprehensively compared the RWGS performance metrics of the MoO$_3$/Mo$_2$N-C catalyst, the passivated MoO$_3$/Mo$_2$N-C catalyst (lacking MoO$_x$), and the Mo$_2$N-C catalyst (lacking the surface MoO$_3$ oxide layer). As shown in Supplementary Fig. 29a–e, the MoO$_3$/Mo$_2$N-C catalyst exhibits the highest CO$_2$ conversion and CO formation rate, lowest CO$_2$ reaction order and activation energy. These results suggest that MoO$_x$, rather than MoO$_3$, serves as the primary active species in the catalytic system. In contrast, the Mo$_2$N-C catalyst demonstrates the lowest H$_2$ reaction order, indicating the strongest H$_2$ activation capability when Mo$_2$N is fully exposed. The CO$_2$ and H$_2$ reaction orders, as well as the activation energy, reflect the dependence of the catalytic reaction on CO$_2$ concentration, H$_2$ concentration, and temperature, respectively. Supplementary Fig. 29f visually summarizes the roles of Mo$_2$N and MoO$_x$ in the RWGS reaction mechanism, demonstrating that the exposure of Mo$_2$N reduces the reaction's dependence on H$_2$ concentration, while MoO$_x$ significantly reduces its dependence on both CO$_2$ concentration and temperature.

Additionally, we synthesized unsupported Mo$_2$N and Mo$_2$C via ammonia and methane reduction methods (SEM, Supplementary Fig. 30), with XRD characterization confirming perfect alignment of all lattice diffraction peaks with standard Mo$_2$N and Mo$_2$C reference patterns. Subsequent Raman analysis reveals oxide layers on the unsupported Mo$_2$N and Mo$_2$C surfaces, where Mo$_2$N exhibits stronger MoO$_3$ Raman signals than Mo$_2$C, which is consistent with trends observed for MoO$_3$/Mo$_2$N-C and MoO$_3$/MoO$_2$-C catalysts

(Supplementary Fig. 31). Furthermore, CO$_2$ conversion tests in RWGS reactions before and after H$_2$ treatment demonstrate unchanged performance for Mo$_2$C but enhanced conversion rates for Mo$_2$N, unequivocally establishing identical surface reconstruction behavior to our catalysts (Supplementary Fig. 32).

To get further insight into the dynamic surface transformations of the catalysts throughout the RWGS reaction, in situ Raman spectroscopy was employed over a temperature range of 100–500 °C, as shown in Fig. 6a–c. As illustrated in Fig. 6a for MoO$_3$/Mo$_2$N-C, the surface MoO$_3$ is reduced already at 100 °C, as only peaks corresponding to MoO$_x$ are detectable across the entire temperature range. The in-situ formation of MoO$_x$ species introduces many oxygen vacancies, which are postulated to be the predominant active sites catalyzing the RWGS reaction. For the MoO$_3$/MoO$_2$-C catalyst, the Raman spectra depicted in Fig. 6b indicate that the surface MoO$_3$ structure has already been reduced to MoO$_2$ at 100 °C. The characteristic peaks attributed to MoO$_2$ diminish with further temperature increase to 500 °C, suggesting that the surface MoO$_2$ is likely to constitute the principal active phase in the MoO$_3$/MoO$_2$-C catalyst. Meanwhile, surface MoO$_3$ in the MoO$_3$/Mo$_2$C-C catalyst demonstrates remarkable stability throughout the RWGS reaction from 100 to 500 °C, (Fig. 6c). Note that due to the small amount of MoO$_3$ on the surface and the increasing blackbody radiation with temperature, the peaks of MoO$_3$ seem to disappear at ~300 °C, however no peaks of MoO$_x$ or MoO$_2$ can be identified instead. To further support our findings, commercial MoO$_3$ was examined under identical

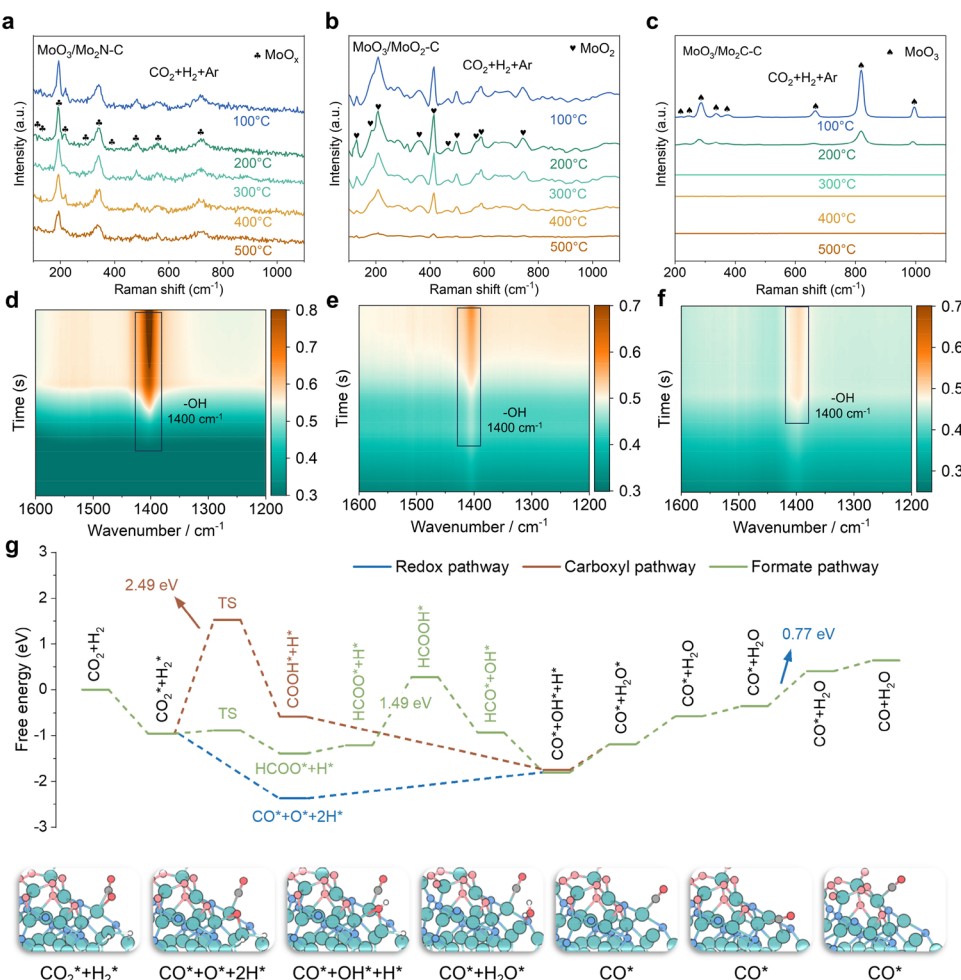

**Fig. 6 | Mechanism exploration of molybdenum-based catalysts under the RWGS reaction. a–c** In-situ Raman spectroscopy results of $MoO_3/Mo_2N$-C, $MoO_3/MoO_2$-C, and $MoO_3/Mo_2C$-C under RWGS reaction conditions. **d–f** In-situ DRIFT spectra of the RWGS reaction over $MoO_3/Mo_2N$-C, $MoO_3/MoO_2$-C, and $MoO_3/Mo_2C$-C at 25–500 °C (Pretreatment condition: 500 °C in $H_2$ diluted in Ar stream (1 mL min$^{-1}$ for $H_2$ and 9 mL min$^{-1}$ for Ar) for 1 h. Reaction conditions: 12 % $H_2$ and 4 %

$CO_2$ in Ar ($H_2/CO_2$ molar ratio of 3) at 10 mL min$^{-1}$. **g** Free energy profiles of the three reaction pathways (redox, carboxyl, and formate) on $MoO_3/Mo_2N$-C. The configurations of intermediates in the redox route are displayed at the bottom, while those in the other two pathways (carboxyl and formate) are shown in Supplementary Fig. 38.

conditions. Supplementary Fig. 33 reveals that the $MoO_3$ surface remains unaltered even at the elevated temperature of 500 °C during the RWGS reaction, with no signals of $MoO_2$ or $MoO_x$. These observations suggest that the reduction of $MoO_3$ is confined to the heterostructured surface, with the nature of the reduced species closely linked to the specific substrate. The $MoO_3/Mo_2N$ heterostructure proves to be most suitable for the formation of a highly active $MoO_x$ surface, resulting in the superior performance of the $MoO_3/Mo_2N$-C catalyst. This insight into the substrate-dependent reducibility and surface stability of $MoO_3$-based catalysts provides a compelling rationale for the observed variations in catalytic activity and selectivity in the RWGS reaction.

Building upon our investigation, in-situ diffuse reflectance infrared Fourier transform spectroscopy (DRIFTS) was employed to discern the intermediate species present on the $MoO_3/MoO_2$-C, $MoO_3/Mo_2N$-C, and $MoO_3/Mo_2C$-C catalysts throughout the RWGS reaction within a temperature range of 50–500 °C. The DRIFTS spectra, as depicted in Fig. 6d–f and Supplementary Figs. 34–36, exhibit a prominent peak at 1400 cm$^{-1}$, which is attributed to the vibration of hydroxyl groups generated from the activation of $H_2$ on the catalyst surfaces[36,37]. Moreover, the hydroxyl vibrational peak at 1400 cm$^{-1}$ is present throughout the entire RWGS reaction process for all three catalysts. This observation is essential, as it suggests that the $MoO_3/MoO_2$-C,

$MoO_3/Mo_2N$-C, and $MoO_3/Mo_2C$-C catalysts predominantly adhere to a redox mechanism, diverging from the traditional associative mechanism. In the redox mechanism, $CO_2$ is adsorbed onto the catalyst surface, where it undergoes direct C=O bond cleavage under the catalyst's influence to yield CO and an oxygen species O* adsorbed on the catalyst surface. The O* is then reduced by $H_2$ to form $H_2O$, which, after desorption, initiates a new cycle. Consequently, this alternative pathway circumvents the formation of substantial bicarbonate intermediates, thereby culminating in exceptional selectivity for the RWGS reaction. The DRIFTS findings, in conjunction with the Raman spectroscopy results, corroborate the substrate-dependent reducibility and active site formation on the $MoO_3/Mo_2N$-C catalyst, which is crucial for its superior catalytic performance. Isotope labeling experiments have been employed to trace the origin of reactant components. We utilized a feed gas mixture of 2.5% $^{13}CO_2$ + 97.5% Ar and employed in situ mass spectrometry to distinguish the carbon source of the CO product. As shown in Supplementary Fig. 37, when $^{13}CO_2$ is pulsed over the $MoO_3/Mo_2C$-C catalyst in the absence of $H_2$, the C=O bond in $^{13}CO_2$ cleave, generating $^{13}CO$; no $^{12}CO$ is observed. This confirms that neither the surface $MoO_3$ layer nor the bulk $Mo_2C$ in the $MoO_3/Mo_2C$-C catalyst underwent a carbon exchange mechanism during the RWGS reaction. Consequently, all carbon in the produced CO originates exclusively from $CO_2$.

Using first-principles calculations, we further explored the catalytic reaction mechanism from an atomic-scale perspective. Here, both the redox and the associative (including the carboxyl and the formate pathways) mechanisms were considered (Fig. 6g and Supplementary Fig. 38). For the associative mechanism, the hydrogenation of $CO_2$ on $MoO_3$/$Mo_2$N-C is more hindered, especially for the carboxyl pathway with a significantly higher kinetic energy barrier of 2.49 eV, which limits the subsequent reaction. For the formate pathway, the formation of HCOOH* intermediate is hindered due to a higher thermodynamic energy barrier of 1.49 eV. Regarding the redox mechanism, computational simulations reveal that $CO_2$* tends to form CO* and O* spontaneously without any kinetic barrier, and the subsequent reaction process shows mild energy changes with a much lower rate-determining step energy barrier of only 0.77 eV. Thus, compared to the associative mechanism, the RWGS reaction is more likely to follow the redox pathway on $MoO_x$ formed on the $Mo_2$N surface, aligning with experimental results. Furthermore, through a detailed analysis of the redox pathways, we systematically investigated the transformation and desorption pathways of key intermediates (O, OH, and CO). As illustrated in Supplementary Fig. 39, $CO_2$ and $H_2$ undergo catalytic activation on the surface to form adsorbed intermediates: CO*, O*, and H*. The reaction proceeds via the sequential hydrogenation of the O* intermediate, first forming OH*, followed by $H_2$O*, which ultimately desorbs. The remaining CO* intermediate subsequently desorbs, completing the catalytic cycle and regenerating the active surface. Notably, a comparative analysis of potential pathways reveals that this mechanism possesses the most favorable thermodynamics, with an energy barrier of only 0.77 eV, significantly lower than those of other pathways (0.97, 0.97, and 1.16 eV).

## Discussion

In this study, we address the complex phenomenon of the surface sensitivity and dynamic nature of catalyst surface remodeling under high temperatures, which is essential for understanding and finally enhancing the catalytic efficiency of heterogeneous catalysts. While the interactions between active materials and substrates in supported catalyst systems have been extensively studied, the impact of substrate type on surface reconstruction in Mo-based catalysts has remained largely unexplored. Our work sheds new light on the intricate chemistry of catalyst surface reconstruction and its dependence on substrate type, revealing a direct correlation between the reducibility of surface $MoO_3$ and the nature of the underlying Mo-based materials ($Mo_2$N, $MoO_2$, and $Mo_2$C). Comprehensive in-situ characterization techniques and theoretical analyses reveal that the $MoO_3$ layer on $MoO_2$ and $Mo_2$N is in-situ reduced to $MoO_2$ or metastable $MoO_x$ (2 < x < 3), respectively, but not reduced at all on $Mo_2$C, during the catalysis process. We demonstrate that the surface stress exerted by different substrates on $MoO_3$ leads to different oxidation states, with the $MoO_3$/$Mo_2$N-C catalyst exhibiting an optimal level of reducibility that stabilizes metastable $MoO_x$ active sites, which shows an unprecedented CO yield of up to 48.2%, approaching the equilibrium conversion limit, the CO formation rate of $8.23 \times 10^{-5}$ $mol_{CO}$ $g_{cat}^{-1}$ $s^{-1}$, and up to 99% CO selectivity under 500 °C. These metastable species are crucial for the efficient activation of $CO_2$; meanwhile, $Mo_2$N, which possesses noble metal-like properties in $H_2$ dissociation, facilitates a direct redox pathway on the $MoO_3$/$Mo_2$N-C catalyst. This pathway is responsible for the exceptional selectivity and stability observed in high-temperature RWGS reactions. Our findings not only deepen the understanding of how heterostructure interfaces influence catalytic performance but also offer valuable guidelines for developing efficient heterogeneous catalysts tailored for various catalytic reactions. By elucidating the interplay between substrate type and surface reconstruction, we pave the way for the design of catalysts with targeted properties, poised to unlock new possibilities in catalysis.

## Methods

### Synthesis of the organic-polyoxometalate c$MoO_3$/crystals

Ammonium molybdate tetrahydrate (($NH_4$)$_6$$Mo_7$$O_{24}$·$4H_2O$) served as the Mo−POM precursor, with p-phenylenediamine as the organic ligand in an aqueous medium. In a typical procedure, 1.08 g of p-phenylenediamine was dissolved in 50 mL of deionized water, followed by the dropwise addition of 2.48 g of ($NH_4$)$_6$$Mo_7$$O_{24}$·$4H_2O$ dissolved in another 50 mL of water. Subsequently, 34 mL of 1 M HCl was introduced to modulate the morphology of the assembled co-crystals. The mixture was reacted for 3 h, after which the solid product was collected via filtration and thoroughly washed with water and ethanol.

### Synthesis of molybdenum-based catalysts

The as-synthesized precursors were carbonized under an Ar atmosphere at different temperatures−500 °C, 700 °C, and 900 °C−using a heating rate of 5 °C·$min^{-1}$ and a 2 h dwell time. After cooling to room temperature, the resulting materials were passivated in a 1% $O_2$/Ar flow for 12 h to form a protective surface $MoO_3$ layer while avoiding bulk oxidation of the $Mo_2$N and $Mo_2$C phases. The final catalysts are denoted as $MoO_3$/$MoO_2$-C (500 °C), $MoO_3$/$Mo_2$N-C (700 °C), and $MoO_3$/$Mo_2$C-C (900 °C). Unless otherwise specified, these designations refer to samples carbonized at the respective temperatures.

### Catalytic measurements

The $CO_2$ hydrogenation activity for the RWGS reaction was assessed in a fixed-bed reactor operating at atmospheric pressure. In each test, 100 mg of catalyst (sieve fraction) was diluted with 100 mg of inert $SiO_2$ and loaded into a quartz tube. Prior to reaction, the catalyst was pretreated in a 15% $H_2$/Ar stream (100 mL·$min^{-1}$) at 500 °C for 1 h. After cooling to room temperature, the feed was switched to the RWGS reaction mixture (23% $CO_2$, 69% $H_2$, 8% Ar) at a total flow rate of 100 mL·$min^{-1}$. The reaction was maintained for 60 min at each temperature to reach steady state before product analysis. The system pressure was maintained at 1 bar using a back-pressure regulator. Effluent gases were analyzed online using an Agilent 7890B gas chromatograph equipped with two TCDs and one FID. Argon served as the internal standard for gas flow quantification. $CO_2$ conversion was calculated based on the following expression:

$$X_{CO_2}(\%) = \frac{n_{in}^{CO_2} - n_{out}^{CO_2}}{n_{in}^{CO2}} \times 100\% = \left(1 - \frac{\frac{A_{out}^{CO_2}}{A_{out}^{Ar}}}{\frac{A_{in}^{CO_2}}{A_{in}^{Ar}}}\right) \times 100\% \qquad (1)$$

where $n_{in}^{CO_2}$ is the concentration of $CO_2$ in the reaction stream, and $n_{out}^{CO_2}$ is the concentration of $CO_2$ in the outlet gas. $A_{in}^{CO_2}$ and $A_{in}^{Ar}$ refer to the chromatographic peak area of $CO_2$ and Ar in the inlet gas, respectively, and $A_{out}^{CO_2}$ and $A_{out}^{Ar}$ refer to the chromatographic peak area of $CO_2$ and Ar in the outlet gas, respectively. The chromatographic peak area of each component is proportional to the concentration of each component.

CO selectivity was calculated by the following equations:

$$S_{CO}(\%) = \frac{n_{out}^{CO}}{n_{out}^{CO} + n_{out}^{CH_4}} \times 100\% = \frac{A_{out}^{CO} \times f_{CO/Ar}}{A_{out}^{CH_4} \times f_{CH_4/Ar} + A_{out}^{CO} \times f_{CO/Ar}} \times 100\%$$

$$(2)$$

where $n_{in}^{CO}$ and $n_{in}^{CH_4}$ refer to the concentration of CO and $CH_4$ in the outlet gas, respectively. $f_{CO/Ar}$ and $f_{CH4/Ar}$ are relative correction factors of CO to Ar and $CH_4$ to Ar, respectively, which are determined by the calibrating gas. $A_{out}^{CO}$ and $A_{out}^{CH_4}$ are the chromatographic peak areas of CO and $CH_4$ detected by the FID in the outlet gas.

The carbon balance was calculated as:

$$C_{balance}(\%) = \frac{n_{out}^{CO_2} + n_{out}^{CO} + n_{out}^{CH_4}}{n_{in}^{CO}} \times 100\% \qquad (3)$$

where $n_{in}^{CO}$ is the concentration of CO in the reaction stream, $n_{out}^{CO_2}$, $n_{out}^{CO_2}$, and $n_{out}^{CO_2}$ are the concentration of $CO_2$, CO, and $CH_4$ in the outlet gas.

## Data availability

All data supporting the findings of this study are available within this article and Supplementary Information or from the corresponding author upon request. The data generated in this study are provided in the Supplementary Information/Source Data file. Source data are provided with this paper.

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

## Acknowledgements

This work was financially supported by the National Natural Science Foundation of China (Nos. 52273269 [S.L.]; 52473278 [S.L.]) and the Sichuan Science and Technology Program (Nos. 2023YFH0027 [S.L.], 2023YFH0008 [C.C.]). We acknowledge the financial support from the Fundamental Research Funds for the Central Universities. We gratefully acknowledge Dr. Mi Zhou and Dr. Chao He at Sichuan University for their experimental assistance and analytical support. We thank the Analytical & Testing Center of Sichuan University for their assistance on XPS work performed by Yunfei Tian, Suilin Liu and Shuguang Yan.

## Author contributions

Y.F.F. and Z.Y.X. contributed equally to this work. Y.F.F., Z.Y.X., C.C., A.T. and S.L. conceived the idea and designed the experiments. D.P.Y., Y.T. and J.N. helped with the RWGS experiments and results analysis. D.P.Y., B.Y. and T.M. assisted with the Fig. production and experiment design. Y.F.F., Z.Y.X., C.C., A.T. and S.L. designed and performed the theoretical calculation. Y.F.F., Z.Y.X., C.C., S.L. and A.T. wrote and edited the manuscript. C.C., A.T. and S.L. supervised the whole project. All authors discussed the results and commented on the manuscript.

## Funding

## Competing interests

The authors declare no competing interests.
