## [Transparent Peer Review file · Nature Communications]

Unraveling surface sensitivity for generating metastable active site in molybdenum-based catalysts for CO₂ hydrogenation

Corresponding Author: Professor Arne Thomas

Version 0:

Reviewer comments:

Reviewer #1

(Remarks to the Author)

This manuscript by Li et al. investigated the effect of three varied support for MoO₃ and used the formed composite catalysts for CO₂ hydrogenation. A metastable MoO_x species on MoO₃/Mo₂N-C was formed and showed good CO yield up to 48.2 % with a CO formation rate of 8.23×10^{-5} molCO gcat⁻¹ s⁻¹ as well as 99 % CO selectivity under 500 °C. In situ characterization and theoretical analyses were adopted to confirm the hypothesis. The results are interesting. Before possible consideration of publication in Nat. Commun., the authors need to address the following issues.

- 1) Scheme 1 is not much related to the concept of this work and can be removed.
- 2) What does the color stand for in Fig. 1a.
- 3) The authors claimed the weakest interaction between the surface MoO₃ and Mo₂O-C support and thus stability of the MoO₃ against reduction. This is difficult to understand. Normally weaker interaction leads to easier reduction.
- 4) The models used for DFT calculations need to be clarified in the DFT parts. How to ensure the accuracy of the models applied? Are they consistent with experimental characterization results?
- 5) The XPS results indicated the largest BE shift for MoO₃/MoO₂-C. Does this suggest the strongest interaction between the surface MoO₃ and MoO₂-C? If so, why did the composite exhibit lower CO₂ adsorption than MoO₃/Mo₂N-C?
- 6) Did the support Mo₂N, MoO₂, Mo₂C remain stability during H₂ treatment?
- 7) The authors ascribed the formation of oxygen vacancies in MoO₃ after H₂ treatment, however, did MoO₂ and Mo₂O also result in oxygen vacancies during the H₂ treatment?
- 8) What are the respective CO₂ hydrogenation results for the three bare supports, Mo₂N-C, MoO₂-C, and Mo₂C-C ?
- 9) For accuracy, standard deviations or errors need to be added for the catalytic data.
- 10) Did the formed *OH react with *H to form H₂O followed by desorption of CO or the other way round? Did the authors calculate these?
- 11) Long-term stability tests need to be performed and added.

Reviewer #2

(Remarks to the Author)

The investigation of catalytic surfaces in Mo₂C and Mo₂N systems for the reverse water-gas shift (RWGS) reaction represents a scientifically significant research direction. However, their catalytic surfaces are highly complex, involving MoO_x, MoOC_x or MoON_x, as well as Mo₂C and Mo₂N. Therefore, the authors' limited discussion on the influence of a single MoO_x species on catalytic performance is insufficiently credible and inadequate. Consequently, the findings of this study are not strong enough to warrant publication in the Nature Communications. Additionally, there are some issues and suggestions which should be reviewed by the authors in order to have a better article.

- (1) In addition to investigating the influence of molybdenum species on the catalyst surface on activity, it is also necessary to examine the effects of catalyst particle size, specific surface area, and carbon content.
- (2) Given the complicated nature of the current catalyst system containing multiple elements (C, N, O), we strongly suggest the authors employ a simplified unsupported system. Preparation of unsupported Mo₂N and Mo₂C through conventional ammonia/methane reduction method would enable more conclusive investigation of surface molybdenum species' catalytic

behavior.

(3) Since the redox pathway likely involves the $\text{MoOCx} \leftrightarrow \text{Mo}_2\text{C}$ —a well-known process—the authors should expand their mechanistic discussion to include this aspect.

(4) The weak and ambiguous signals in situ Raman data (Fig.5) do not provide convincing evidence for the authors' conclusions.

(5) The manuscript contains some minor writing errors that require correction, particularly in Page 8 (lines 144-145) and Figures 4d and 4e.

Reviewer #3

(Remarks to the Author)

Molybdenum-based catalysts are investigated for CO_2 hydrogenation. In particular, the RWGS is the thermal catalytic process considered to explore. The analysis of the surface reconstruction and its consequences in catalytic performance. Analyzing the surface reconstruction is very important because it may conditionate the performance of the systems with the time, even promote the passivation. The authors emphasize that this study constitutes the first time the formation of MoOx sites is reported and this could pave a way to design heterogeneous catalysts by constructing highly active metal oxides and adjusting the reducibility of substrates.

Mo_2N and Mo_2C are the substrates that (if understood correctly) form coherent 2D nanosheet nanostructures. What is the thickness of these systems? One may wonder about these are 3D carbides or MXenes with a certain thickness. The structural analysis shows Mo_2N (220), MoO_2 (-211), and Mo_2C (110), but the schematic illustration (Fig 1 a) shows the Mo_2X (111) ($\text{X}=\text{C}, \text{N}$) passivated by MoOx .

The connection between experiments and theory is fantastic. The computational strategy followed is reasonable and well-considered. It is important to see that redox mechanism is the most favorable respect to other routes. This aspect has been also observed in MXenes in studies published by Morales-García et al. It is important to see that the authors explore the potential energy landscape of the reaction including thermodynamic and kinetics aspects. They are now in a privilege situation of move and step further performing microkinetic simulations to get the reaction rates and TOFs values and compare directly with their experiments (Fig, 4 g, h, and i). I strongly recommend to add this multiscale analysis and I have curiosity the matching with experiments.

Version 1:

Reviewer comments:

Reviewer #1

(Remarks to the Author)

The authors have properly revised their manuscript according to the reviewers' comments. It can now be considered for publication.

Reviewer #2

(Remarks to the Author)

The authors emphasized that activity analysis indicates the formation of metastable MoOx species on $\text{MoO}_3/\text{Mo}_2\text{N-C}$, which contributes to its unprecedented RWGS performance. However, XPS results show that Mo_2N species remains the dominant component on the surface. Thus, the respective catalytic roles of MoOx and Mo_2N remain unclear. I strongly recommend that the authors further compare the RWGS activity over the passivated $\text{MoO}_3/\text{Mo}_2\text{N-C}$ catalyst, the H_2 -pretreated $\text{MoO}_3/\text{Mo}_2\text{N-C}$ catalyst, and the fresh (non-passivated) $\text{MoO}_3/\text{Mo}_2\text{N-C}$ sample to clarify the true catalytic nature of the Mo_2N catalyst.

Reviewer #3

(Remarks to the Author)

I appreciate the effort made by the authors in addressing the Reviewer's concerns. The revised version of the manuscript is now suited for publication

Version 2:

Reviewer comments:

Reviewer #2

(Remarks to the Author)

The authors' efforts in comprehensively responding to the reviewers' feedback are commendable. With all concerns now satisfactorily resolved, I recommend the revised manuscript for acceptance.

Point-by-point response to the comments of the reviewers for the publication “Unraveling surface sensitivity for generating metastable active site in molybdenum-based catalysts for CO₂ hydrogenation”, manuscript ID: NCOMMS-25-25695T.

REVIEWER COMMENTS

Reviewer #1 (Remarks to the Author):

“This manuscript by Li et al. investigated the effect of three varied support for MoO₃ and used the formed composite catalysts for CO₂ hydrogenation. A metastable MoO_x species on MoO₃/Mo₂N-C was formed and showed good CO yield up to 48.2 % with a CO formation rate of $8.23 \times 10^{-5} \text{ molCO g}_{\text{cat}}^{-1} \text{ s}^{-1}$ as well as 99 % CO selectivity under 500 °C. In situ characterization and theoretical analyses were adopted to confirm the hypothesis. The results are interesting. Before possible consideration of publication in Nat. Commun., the authors need to address the following issues.”

Comment 1: Scheme 1 is not much related to the concept of this work and can be removed.

Response to comment 1: Thank you for your suggestion regarding Scheme 1. We agree that streamlining the manuscript enhances conceptual focus. We have removed Scheme 1 and related contents.

Comment 2: What does the color stand for in Fig. 1a.

Response to comment 2: We thank the reviewer for pointing out this issue, which was indeed unclear. We have added annotations and explanations at the relevant locations in Fig. 1a. The corresponding details have also been incorporated into the revised manuscript, as shown below:

Fig. 1 Characterization of the heterostructured molybdenum-based catalysts. a) Schematic illustration of the catalyst's surface structure. Light blue: molybdenum atoms; dark blue: nitrogen atoms; pink: oxygen atoms; gray: carbon atoms.

Comment 3: *The authors claimed the weakest interaction between the surface MoO_3 and $\text{Mo}_2\text{C-C}$ support and thus stability of the MoO_3 against reduction. This is difficult to understand. Normally weaker interaction leads to easier reduction.*

Response to comment 3: Thank you for this comment. We agree with the reviewer that, normally, a weaker interaction with the oxidative substrate might lead to easier reduction. However, here the interaction is with a reductive substrate, Mo_2C , which might bring the opposite result. Previous studies also demonstrated that the interaction between MoO_3 and a reductive substrate facilitates the reduction of MoO_3 (*J. Am. Chem. Soc.* 2020, 142, 31, 13362–13371). Furthermore, the in-situ Raman spectroscopy of pure MoO_3 corroborates this finding: under H_2 atmosphere, pure MoO_3 exhibits no reduction, whereas MoO_3 supported on $\text{Mo}_2\text{N-C}$ ($\text{MoO}_3/\text{Mo}_2\text{N-C}$) is easily reduced to MoO_x under identical conditions. The interactions between the MoO_3 and different substrates can be revealed and evaluated by DFT calculations. The computational results demonstrate that distinct interaction strengths induce variations in Mo-O bond lengths within the surface oxide layer (Fig. 2f). The weakest interaction between MoO_3 and $\text{Mo}_2\text{C-C}$ results in the shortest Mo-O bond length in $\text{MoO}_3/\text{Mo}_2\text{C-C}$, rendering these bonds most difficult to cleave. Consequently, among the three catalysts, $\text{MoO}_3/\text{Mo}_2\text{C-C}$ exhibits the lowest reducibility. As shown in Fig. 2g, projected density of states (pDOS) analysis reveals divergent distributions of electronic states near the Fermi level in the MoO_3 layer across different supports. The catalyst with the strongest interaction ($\text{MoO}_3/\text{Mo}_2\text{N-C}$) exhibits electronic states at the Fermi level, whereas the system with the weakest interaction ($\text{MoO}_3/\text{Mo}_2\text{C-C}$) shows negligible electronic states at this energy. This suggests that electron transfer is impeded in the weakly interacting $\text{MoO}_3/\text{Mo}_2\text{C-C}$ system.

To make this point clearer, we have expanded our discussion in the revised manuscript to include additional descriptions of how interactions affect the reducibility of surface MoO_3 , as also shown below:

Page 8 in the revised manuscript: “When the substrate (Mo_2N , MoO_2 , Mo_2C) comes into contact with the surface layer (MoO_3), different electronic interactions arise, thereby influencing the properties of the surface layer. The calculation results show that in $\text{MoO}_3/\text{Mo}_2\text{N-C}$, the Mo-O bonds in the surface MoO_3 layer are significantly elongated with a value of 1.98 Å resulting from the strong interaction between Mo_2N and MoO_3 , weakening the strength of the Mo-O bonds; while $\text{MoO}_3/\text{MoO}_2\text{-C}$ and $\text{MoO}_3/\text{Mo}_2\text{C-C}$ exhibit a relatively shorter Mo-O bond length of 1.89 Å and 1.80 Å, respectively (insets in Fig. 2f). Moreover, the pDOS plots in Fig. 2g indicate that the MoO_3 layer in $\text{MoO}_3/\text{Mo}_2\text{N-C}$ exhibits a relatively prominent electronic state near the Fermi level, which facilitates electron transfer during the reaction and promotes the formation of oxygen vacancies, as further supported by the oxygen vacancy formation energy calculations shown in Fig. 2f. However, in $\text{MoO}_3/\text{MoO}_2\text{-C}$ and $\text{MoO}_3/\text{Mo}_2\text{C-C}$, the strong Mo-O bonds and limited surface electron transfer, reflected by their larger band gaps, make the formation of MoO_x species more difficult.”

Supplementary Figure 32. In-situ Raman results of commercial- MoO_3 under the RWGS reaction.

Fig. 2c. In-situ Raman results of MoO₃/Mo₂N-C under H₂ pretreatment. The inset shows schemes of the surface structure of three catalysts in different atmospheres.

Fig. 2f and g. **f)** Calculated formation energy of oxygen vacancy and Mo-O (MoO₃) bond length of different materials. **g)** PDOS analysis of the Mo-O (MoO₃) bond for different materials.

Comment 4: *The models used for DFT calculations need to be clarified in the DFT parts. How to ensure the accuracy of the models applied? Are they consistent with experimental characterization results?*

Response to comment 4: Thank you for this helpful suggestion. It is unequivocally confirmed that the DFT computational models align precisely with experimental characterization results. To ensure accuracy, the models used for DFT calculations were constructed based on comprehensive experimental characterizations. Specifically, HAADF-STEM lattice spacing and XRD patterns identify the substrate composition of the three catalysts, which are Mo₂N, MoO₂, and Mo₂C,

respectively (Fig. R1). Subsequently, Raman spectra confirmed the presence of MoO₃ overlayers on all catalysts, while no MoO₃ diffraction peaks or lattice fringes were detected in HAADF-STEM/XRD data, indicating an ultrathin surface MoO₃ layer. Accordingly, we established the DFT models with a thin layer of MoO₃ coated on each substrate surface (Fig. R2). Finally, distinct surface reconstructions during RWGS observed via in situ Raman spectroscopy guided the construction of DFT models for the catalytic reaction state, accompanied by MoO_x, MoO₂, and MoO₃ on the surface of Mo₂N, MoO₂, and Mo₂C, respectively (Fig. R3). To make this clearer for readers, we have expanded the DFT methodology section to detail this model construction process, as shown below:

Page 5-6 in the revised Supplementary Materials: “Where $E_{(surf)}$ and $E_{(mol)}$ are the energies of substrates and isolated molecules, respectively, and $E_{(surf+mol)}$ represents the energy of the combined systems upon adsorption. This means that a negative E_{ads} value corresponds to exothermic adsorption. The models used in DFT calculations were constructed based on the comprehensive experimental characterization results. First, Mo₂N, MoO₂, and Mo₂C are identified from HAADF-STEM and XRD. Subsequently, Raman spectra further reveal that a few layers of MoO₃ are present on the three catalysts, which are thin enough that no MoO₃ diffraction peaks or lattice fringes are detected in XRD and HAADF-STEM characterization. Finally, different surface reconstructions during the RWGS process were observed by in-situ Raman spectroscopy, which guided the construction of DFT models under catalytic reaction conditions.”

Fig. R1. a1, b1, c1) HAADF-STEM images of MoO₃/Mo₂N-C, MoO₃/MoO₂-C, and MoO₃/Mo₂C-C; a2, b2, c2) false color image of the HAADF-STEM image from a1, b1, c1); d, e, f) XRD patterns of

MoO₃/Mo₂N-C, MoO₃/MoO₂-C, and MoO₃/Mo₂C-C; **g, h, i**) DFT calculation model of the substrates of MoO₃/Mo₂N-C, MoO₃/MoO₂-C, and MoO₃/Mo₂C-C. Light blue: molybdenum atoms; dark blue: nitrogen atoms; pink: oxygen atoms; gray: carbon atoms.

Fig. R2. **a, b, c**) XRD patterns and Raman spectra of MoO₃/Mo₂N-C, MoO₃/MoO₂-C, and MoO₃/Mo₂C-C; **d, e, f**) DFT calculation model of heterojunction between the substrate and surface MoO₃ layer of MoO₃/Mo₂N-C, MoO₃/MoO₂-C, and MoO₃/Mo₂C-C.

Fig. R3. **a, b, c**) In-situ Raman results of MoO₃/Mo₂N-C, MoO₃/MoO₂-C, and MoO₃/Mo₂C-C under H₂ pretreatment; **d, e, f**) DFT calculation model of MoO₃ layer reconstruction on the surface of MoO₃/Mo₂N-C, MoO₃/MoO₂-C, and MoO₃/Mo₂C-C heterojunctions.

Comment 5: *The XPS results indicated the largest BE shift for MoO₃/MoO₂-C. Does this suggest the strongest interaction between the surface MoO₃ and MoO₂-C? If so, why did the composite exhibit lower CO₂ adsorption than MoO₃/Mo₂N-C?*

Response to comment 5: Thank you for your insightful comments and helpful suggestions. We sincerely apologize for the confusion caused by our prior inaccuracies. The presence of Mo⁶⁺ in XPS confirms the existence of surface MoO₃. However, due to inherent differences in Mo valence states among the three catalysts, the valence state of Mo in MoO₂ differs substantially from those in Mo₂N and Mo₂C, resulting in an inevitable elevation of the average Mo state in MoO₂. Importantly, this phenomenon does not reflect the interaction strength between the surface layer and substrate.

Meanwhile, the interaction between the surface MoO₃ and the support does not correlate with CO₂ adsorption capacity. The oxygen vacancies generated by lattice oxygen liberation from surface MoO₃ are the active sites for CO₂ adsorption and activation (*Chem. Sci.*, 2021,12, 9902-9915; *J. Mater. Chem. A*, 2022,10, 10854-10864; *Chem. Eng. J.* 2024, 495, 153333; *Nat. Commun.* 2022, 13, 5800.). This is also in accordance with our DFT calculations results, as shown in Fig. 3a-c, MoO₃/Mo₂N-C, with many oxygen vacancies in the surface MoO_x species, possesses much stronger binding strength for CO₂ (-0.42 eV) compared to MoO₃/MoO₂-C (-0.26 eV) and MoO₃/Mo₂C-C (-0.23 eV), exhibiting the highest CO₂ adsorption capacity. To avoid any misleading on this point, we have optimized the descriptions of the XPS part in the revised manuscript, shown below:

Page 7 in the revised manuscript: “Fig. 2a presents the high-resolution Mo 3d spectra of these catalysts, where MoO₃/Mo₂N-C and MoO₃/Mo₂C-C showed apparent peaks of Mo^{δ+} at 228.8 and 231.9 eV, indicating the existence of Mo₂N and Mo₂C as the main components. In contrast, in MoO₃/MoO₂-C, only the peaks for Mo⁴⁺ (MoO₂, 229.38 and 233.18 eV) and Mo⁶⁺ (MoO₃, 232.72 and 235.83 eV) can be observed, indicating that the stoichiometric Mo oxides are the main component for MoO₃/MoO₂-C. All three catalysts exhibit distinct Mo⁶⁺ peaks, confirming the presence of surface MoO₃ on each material. This result aligns consistently with the Raman characterization data.”

Revised Fig. 2. a) Curve-fitted high-resolution XPS Mo3d spectra of different materials.

Fig. 3 Differential electron density plot, calculated adsorption energy and Bader charge of CO_2^* on **a)** $\text{MoO}_3/\text{Mo}_2\text{N-C}$, **b)** $\text{MoO}_3/\text{MoO}_2\text{-C}$, and **c)** $\text{MoO}_3/\text{Mo}_2\text{C-C}$ (cyan and yellow show charge consumption and accumulation, respectively, the cut-off of the density-difference iso-surface is $0.004 \text{ e}\cdot\text{Bohr}^{-3}$; color codes: light blue, molybdenum; blue, nitrogen; pink, oxygen in lattice; red, oxygen in molecule; gray, carbon in lattice; dark gray, carbon in molecule).

Comment 6: *Did the support Mo_2N , MoO_2 , Mo_2C remain stability during H_2 treatment?*

Response to comment 6: We appreciate the reviewer for for bringing up this important point. Motivated by your question, we conducted supplementary experiments involving H_2 reduction of the three catalysts, followed by immediate XRD characterization to assess potential changes in the

diffraction peaks of the support lattice structures (Mo_2N , MoO_2 , Mo_2C). As shown in **Supplementary Fig. 18a-c**, no structural alterations occur in the Mo_2N , MoO_2 , or Mo_2C supports after H_2 treatment. This convincingly demonstrates that these supports maintain structural stability during H_2 reduction. The corresponding details can be found in the revised manuscript, as also shown below:

Page 12 in the revised manuscript: “At 500 °C under the same WHSV, a CO_2 conversion of 48.2%, 36.5%, 12.6% and CO formation rate of 8.23×10^{-5} , 6.23×10^{-5} , 2.15×10^{-5} $\text{mol}_{\text{CO}} \text{g}_{\text{cat}}^{-1} \text{s}^{-1}$ was obtained for $\text{MoO}_3/\text{Mo}_2\text{N-C}$, $\text{MoO}_3/\text{MoO}_2\text{-C}$, $\text{MoO}_3/\text{Mo}_2\text{C-C}$ catalysts, respectively (Fig. 4d, 4e). Notably, all catalysts exhibited exceptional CO selectivity exceeding 99 % across varying temperatures and space velocities. (Fig. 4f). **In addition, the structure of the supports (Mo_2N , MoO_2 , Mo_2C) of the three catalysts did not change during the H_2 treatment process (Supplementary Fig. 18).”**

Supplementary Fig. 18a, b, c) XRD patterns of $\text{MoO}_3/\text{Mo}_2\text{N-C}$, $\text{MoO}_3/\text{MoO}_2\text{-C}$, and $\text{MoO}_3/\text{Mo}_2\text{C-C}$ before and after H_2 treatment.

Comment 7: *The authors ascribed the formation of oxygen vacancies in MoO_3 after H_2 treatment, however, did MoO_2 and Mo_2O also result in oxygen vacancies during the H_2 treatment?*

Response to comment 7: We are grateful for your insightful comments. MoO_2 exhibits significantly greater resistance to oxygen vacancy formation than MoO_3 due to its metallicity, strong bonding, and stable +4 oxidation state. Mo^{4+} ions possess a d^2 electron configuration, forming stronger mixed ionic-covalent bonds with oxygen ions (*J. Phys. Chem. C* 2016, 120, 8959-8968; *Nanomaterials* 2023, 13, 1319). Consequently, oxygen removal from MoO_2 requires breaking stronger bonds compared to MoO_3 . As shown in **Fig. 2f**, DFT calculation of oxygen vacancy (O_v) formation energies for MoO_3 on various supports and MoO_2 reveals that the O_v formation for MoO_2 (2.59 eV) is significantly more

difficult than for MoO₃ (0.37 eV for MoO₃/Mo₂N-C, 1.25 eV for MoO₃/MoO₂-C, and 1.95 eV for MoO₃/Mo₂C-C). This confirms that MoO₂ generates negligible oxygen vacancies under H₂ treatment. Furthermore, Mo₂O does not exist as a stable binary compound and is absent in our catalyst system. Given its lower oxidation state, Mo₂O might exhibit even greater resistance to oxygen vacancy formation.

Fig. 2f. Calculated oxygen vacancy formation energy on MoO₃/Mo₂N-C, MoO₃/MoO₂-C, MoO₃/Mo₂C-C, and MoO₂.

Comment 8: What are the respective CO₂ hydrogenation results for the three bare supports, Mo₂N-C, MoO₂-C, and Mo₂C-C?

Response to comment 8: We sincerely appreciate this important comment, which helped to further enhance the quality of our manuscript. Since Mo₂N-C, MoO₂-C, and Mo₂C-C inevitably oxidize upon exposure to ambient air, forming a surface MoO₃ layer, the hydrogenation capabilities of these bare supports can only be evaluated through immediate post-synthesis testing within the catalytic reactor.

Accordingly, we performed in situ synthesis of Mo₂N-C (500°C, Ar, 2h) and MoO₂-C (700°C, Ar, 2h) under the same reaction conditions, followed by immediate RWGS reaction to assess their CO₂ hydrogenation capabilities. Thermogravimetric analysis (TGA) determined precise feedstock loading to ensure consistent reaction conditions (WHSV) between bare supports and oxide-coated catalysts (Supplementary Fig. 23). XRD characterization of the synthesized Mo₂N-C and MoO₂-C reveals no

changes in diffraction peaks, confirming preserved lattice structures. This implies that compared to MoO₃/Mo₂N-C and MoO₃/MoO₂-C, these materials differ solely in the absence of surface MoO₃ layers (Supplementary Fig. 24). Analysis of catalytic activity shows that bare supports, Mo₂N-C, and MoO₂-C exhibit reduced CO₂ conversion, with a more pronounced decline for Mo₂N-C (Supplementary Fig. 25a and b). We use the CO formation rate, eliminating discrepancies from catalyst mass variations. The CO formation rate comparison conclusively establishes that surface MoO₃ removal diminishes the CO₂ hydrogenation capability of our catalysts (Supplementary Fig. 25c and d). We supplemented measurements of activation energies and reaction orders for Mo₂N-C and MoO₂-C (Supplementary Fig. 26). The apparent activation energy (E_a) of Mo₂N-C is 47.96 kJ·mol⁻¹, significantly lower than that of MoO₂-C. Concurrently, the H₂ reaction order for Mo₂N-C is markedly lower than for MoO₂-C, aligning with the trend observed in MoO₃/Mo₂N-C and MoO₃/MoO₂-C catalysts. However, Mo₂N-C and MoO₂-C exhibit similar CO₂ reaction orders, diverging from this pattern. This indicates that the surface MoO₃ layer minimally influences the H₂ reaction order but significantly modulates the CO₂ reaction order.

To clarify the role of the oxide layer, we compared reaction orders and activation energies between catalysts with and without surface oxides. As shown in Supplementary Fig. 27, Mo₂N-C and MoO₂-C exhibit substantially higher CO₂ reaction orders than MoO₃/Mo₂N-C and MoO₃/MoO₂-C, demonstrating that MoO₃/Mo₂N-C and MoO₃/MoO₂-C with a surface MoO₃ layer possess superior CO₂ activation capability. Conversely, the oxide layer has minimal impact on H₂ reaction orders, confirming that H₂ activation does not originate from the oxide layer. Here, the CO₂ reaction order reflects the catalyst's sensitivity to CO₂ concentration in RWGS, with a lower value indicating facile CO₂ activation on the catalytic surface. Furthermore, Supplementary Fig. 28 reveals that oxide-free catalysts show elevated activation energies, likely due to impaired CO₂ adsorption and activation after MoO₃ removal, resulting in stronger temperature dependency.

All these corresponding details are discussed and added in the revised manuscript, as shown below:

Page 13 in the revised manuscript: “To compare and understand the specific role of the oxide layer on the catalyst surface on CO₂ hydrogenation, we synthesized Mo₂N-C (500°C, Ar, 2h) and MoO₂-C (700°C, Ar, 2h) in situ under identical reactor conditions, preventing surface MoO₃ formation by

avoiding air exposure, and immediately conducted RWGS reactions. Thermogravimetric analysis (TGA) determined precise feedstock loading to ensure consistent reaction conditions between bare supports and oxide-coated catalysts (Supplementary Fig. 23). Specifically, when the WHSV was 60,000 mL·g_{cat}⁻¹h⁻¹ and the catalyst feed mass was 100 mg, the precursor mass loss at 500°C was 79.7%. When preparing the Mo₂N-C catalyst, the precursor feed mass was 125 mg. At 700°C, the mass loss of the precursor is 56.1%, and the precursor feed mass is 178 mg when preparing the MoO₂-C catalyst. Furthermore, XRD characterization of synthesized Mo₂N-C and MoO₂-C (Supplementary Fig. 24) shows unchanged diffraction peaks, confirming preserved lattice structures, indicating that Mo₂N-C and MoO₂-C differ from MoO₃/Mo₂N-C and MoO₃/MoO₂-C solely by lacking the oxide layer.

Supplementary Fig. 25 shows that the CO₂ conversion rate and CO yield of the bare support Mo₂N-C and MoO₂-C without a surface MoO₃ oxide layer are both lower than those of MoO₃/Mo₂N-C and MoO₃/MoO₂-C. After eliminating minor differences in feedstock quantity through the CO formation rate, it can be observed that the CO formation rate of Mo₂N-C is reduced from 8.26×10^{-5} to 5.19×10^{-5} mol_{CO} g_{cat}⁻¹ s⁻¹ compared to MoO₃/Mo₂N-C, and the CO formation rate for MoO₂-C is reduced from 6.17×10^{-5} to 4.80×10^{-5} mol_{CO} g_{cat}⁻¹ s⁻¹ compared to MoO₃/MoO₂-C. The performance of the bare support Mo₂N-C decreases more significantly. Additionally, as shown in Supplementary Fig. 26, we have included measurements of activation energies and reaction orders for oxide-free Mo₂N-C and MoO₂-C. The apparent activation energy (E_a) of the Mo₂N-C catalyst is determined as 47.96 kJ·mol⁻¹, significantly lower than that of MoO₂-C (65.00 kJ·mol⁻¹). Concurrently, the H₂ reaction order for Mo₂N-C (0.36) is markedly lower than for MoO₂-C (0.61), following the same trend observed for the MoO₃/Mo₂N-C and MoO₃/MoO₂-C catalysts. However, Mo₂N-C and MoO₂-C exhibit similar CO₂ reaction orders (0.64 and 0.71), revealing a distinct pattern. This indicates that the surface MoO₃ layer has minimal impact on the H₂ reaction order but significantly modulates the CO₂ reaction order.

To clarify the role of the oxide layer, we compared CO₂ and H₂ reaction orders and activation energies between catalysts with and without surface oxides. As shown in Supplementary Fig. 27, the CO₂ reaction order for MoO₃/Mo₂N-C is 0.36, while the Mo₂N-C exhibits an increased CO₂ reaction order of 0.64; similarly, MoO₃/MoO₂-C shows a CO₂ reaction order of 0.45 versus MoO₂-C at 0.71. The lower CO₂ reaction orders signify superior CO₂ activation capability in MoO₃/Mo₂N-C and MoO₃/MoO₂-C catalysts, which originates directly from the surface MoO₃ layer. Concurrently, the

negligible difference in H₂ reaction orders confirms that H₂ activation capability is independent of the oxide layer. Notably, oxide-free catalysts exhibit higher activation energies (Supplementary Fig. 28), likely due to impaired CO₂ adsorption and activation without surface MoO₃, resulting in higher temperature dependency.”

Supplementary Fig. 23 Thermogravimetric analysis for the organic-polyoxometalate cMoO₃/crystals.

Supplementary Fig. 24 a) XRD patterns of MoO₃/Mo₂N-C and Mo₂N-C, b) XRD patterns of MoO₃/MoO₂-C and MoO₂-C.

Supplementary Fig. 25 a, b) CO_2 conversion at various temperatures, **c, d)** Catalytic reaction rates at various temperatures. (In a-d, $n = 3$ independent experiments, data are presented as mean values \pm SD)

Supplementary Fig. 26 a) Apparent activation energy (E_a), **b, c)** Kinetic orders of reactants (H_2 and CO_2) for $\text{Mo}_2\text{N-C}$ and $\text{MoO}_2\text{-C}$ catalysts. (In a-c, $n = 3$ independent experiments, data are presented as mean values \pm SD)

Supplementary Fig. 27 a) Kinetic orders of H₂ for MoO₃/Mo₂N-C and Mo₂N-C catalysts, b) Kinetic orders of H₂ and CO₂ for MoO₃/MoO₂-C and MoO₂-C, c) Kinetic orders of CO₂ for MoO₃/Mo₂N-C and Mo₂N-C catalysts, d) Kinetic orders of CO₂ for MoO₃/MoO₂-C and MoO₂-C. (In a-d, n = 3 independent experiments, data are presented as mean values ± SD)

Supplementary Figure 28 a) E_a for MoO₃/Mo₂N-C and Mo₂N-C catalysts. **b)** E_a for MoO₃/MoO₂-C and MoO₂-C catalysts. (In a-b, n = 3 independent experiments, data are presented as mean values ± SD)

Comment 9: *For accuracy, standard deviations or errors need to be added for the catalytic data.*

Response to comment 9: We sincerely appreciate your kind reminder. We have added the standard deviations (SD) to all catalytic datasets. The added SD values demonstrate low dispersion in our catalytic results during testing, indicating that individual observations are closely clustered.

Comment 10: *Did the formed *OH react with *H to form H₂O followed by desorption of CO or the other way round? Did the authors calculate these?*

Response to comment 10: Thank you for your highly valuable comment. The transformation and desorption pathways of key intermediates (O, OH, and CO) have been investigated through a detailed examination of the redox pathway. This analysis provides fundamental mechanistic understanding of the surface reaction processes. As shown in **Supplementary Fig. 38**, the catalytic activation of CO₂ and H₂ on the surface generates the adsorbed intermediates: CO*, O*, and H*. The reaction proceeds through sequential hydrogenation of the O* intermediate, first forming OH* and subsequently H₂O*, which eventually desorbs. The remaining CO* intermediate then desorbs, completing the catalytic cycle and regenerating the active surface. Notably, comparative analysis of the potential pathways demonstrates that this mechanism possesses the most favorable thermodynamics, with an energy barrier of only 0.77 eV, significantly lower than the other pathways (0.97 eV, 0.97 eV, and 1.16 eV). The corresponding details have been added in the revised manuscript, shown below:

Page 17 in the revised manuscript: “Furthermore, through a detailed analysis of the redox pathways, we systematically investigated the transformation and desorption pathways of key intermediates (O, OH, and CO). As illustrated in **Supplementary Fig. 38**, CO₂ and H₂ undergo catalytic activation on the surface to form adsorbed intermediates: CO*, O*, and H*. The reaction proceeds via the sequential hydrogenation of the O* intermediate, first forming OH*, followed by H₂O*, which ultimately desorbs. The remaining CO* intermediate subsequently desorbs, completing the catalytic cycle and

regenerating the active surface. Notably, a comparative analysis of potential pathways reveals that this mechanism possesses the most favorable thermodynamics, with an energy barrier of only 0.77 eV, significantly lower than those of other pathways (0.97, 0.97, and 1.16 eV).”

Supplementary Figure 38. Free energy profiles of the redox reaction pathways on MoO₃/Mo₂N-C.

Comment 11: Long-term stability tests need to be performed and added.

Response to comment 11: We are grateful for this important suggestion. We conducted a 200-hour long-term stability test. The results demonstrate that the MoO₃/Mo₂N-C catalyst exhibits exceptional stability, retaining over 96.7% of its initial activity after the 200-hour evaluation (Supplementary Fig. 19). The corresponding data and discussion have been added in the revised manuscript, as shown below:

Page 12 in the revised manuscript: “We conducted stability tests on the best-performing catalysts. As shown in Supplementary Fig. 19, the MoO₃/Mo₂N-C catalyst exhibited excellent stability, maintaining more than 96.7% of its initial activity after the 200-hour evaluation.”

Supplementary Figure 19. Long-term stability test of MoO₃/Mo₂N-C under 60,000 mL g_{cat}⁻¹h⁻¹. (n = 3 independent experiments, data are presented as mean values ± SD)

Reviewer #2 (Remarks to the Author):

“The investigation of catalytic surfaces in Mo₂C and Mo₂N systems for the reverse water-gas shift (RWGS) reaction represents a scientifically significant research direction. However, their catalytic surfaces are highly complex, involving MoO_x, MoOC_x or MoON_x, as well as Mo₂C and Mo₂N. Therefore, the authors' limited discussion on the influence of a single MoO_x species on catalytic performance is insufficiently credible and inadequate. Consequently, the findings of this study are not strong enough to warrant publication in the Nature Communications. Additionally, there are some issues and suggestions which should be reviewed by the authors in order to have a better article.”

Comment 1: In addition to investigating the influence of molybdenum species on the catalyst surface on activity, it is also necessary to examine the effects of catalyst particle size, specific surface area, and carbon content.

Response to comment 1: Thanks for your important comments on improving the quality of our manuscript. To explicitly characterize catalyst particle size, we conducted statistical analysis of particle dimensions for MoO₃/Mo₂N-C, MoO₃/MoO₂-C, and MoO₃/Mo₂C-C. The average sizes are 1.62 ± 0.49 nm, 1.68 ± 0.61 nm, and 1.99 ± 1.30 nm, respectively, indicating no significant differences (Supplementary Fig. 20). Brunauer-Emmett-Teller (BET) surface areas derived from nitrogen adsorption-desorption isotherms are 2.33, 3.05, and 2.31 m² g⁻¹ for MoO₃/Mo₂N-C, MoO₃/MoO₂-C, and MoO₃/Mo₂C-C, respectively, with average pore diameters of 2.60, 2.38, and 4.08 nm (Supplementary Fig. 21). These results confirm roughly the same pore sizes and specific surface areas across all three catalysts. XPS analysis (Supplementary Fig. 22) provides roughly elemental compositions (at.%), although direct comparison of carbon content is complicated by contributions from both the carbon layer and the molybdenum carbide in MoO₃/Mo₂C-C. To resolve this, we included pure Mo₂C XPS as a control, estimating the carbon layer content in each catalyst. As shown in Supplementary Fig. 22b, pure Mo₂C contains 44.4 at.% carbon, implying the carbon layer in MoO₃/Mo₂C-C constitutes 31.6 at.% carbon, which is similar to MoO₃/Mo₂N-C and MoO₃/MoO₂-C. In summary, while investigating the impact of surface Mo species on activity, we maintained constant particle size, surface area, and carbon content as controlled variables. The corresponding details have been added in the revised manuscript, shown below:

Page 12 in the revised manuscript: “To investigate the impact of surface molybdenum species reconstruction on activity in our catalysts, we controlled variables including particle size, specific surface area, and carbon content across the three catalysts. Since all catalysts were obtained from the same precursor via different calcination temperatures, their physical properties (e.g., particle size, surface area) are highly similar. As shown in Supplementary Fig. 20 and 21, particle size distribution histograms and Brunauer-Emmett-Teller (BET) surface areas calculated from nitrogen adsorption-desorption isotherms confirm that MoO₃/Mo₂N-C, MoO₃/MoO₂-C, and MoO₃/Mo₂C-C exhibit comparable particle sizes (1.62 ± 0.49 nm, 1.68 ± 0.61 nm, and 1.99 ± 1.30 nm) and specific surface areas (2.60 m²·g⁻¹, 2.38 m²·g⁻¹, and 4.08 m²·g⁻¹). Furthermore, XPS analysis (Supplementary Fig. 22) quantified the carbon layer proportion in each catalyst. Due to carbon contributions from both the carbon layer and molybdenum carbide in MoO₃/Mo₂C-C, we included pure Mo₂C XPS as a reference, estimating the carbon content within the carbon layer of MoO₃/Mo₂C-C. As shown in Supplementary

Fig. 22b, the carbon content in pure Mo₂C is ~44.4 at.%. Therefore, the carbon content in the carbon layer of the MoO₃/Mo₂C-C catalyst can be estimated at approximately 31.6 at.%, indicating that the carbon layer content is similar across all three catalysts.”

Supplementary Figure 20 STEM images-derived statistical analysis of the crystal sizes of a) MoO₃/Mo₂N-C, b) MoO₃/MoO₂-C, and c) MoO₃/Mo₂C-C.

Supplementary Figure 21 a) N₂ adsorption/desorption isotherms of MoO₃/Mo₂N-C, MoO₃/MoO₂-C, and MoO₃/Mo₂C-C, b) Pore size distribution of MoO₃/Mo₂N-C, MoO₃/MoO₂-C, and MoO₃/Mo₂C-C.

Supplementary Figure 22. a) XPS survey spectra and elemental content for MoO₃/Mo₂N-C, MoO₃/MoO₂-C, and MoO₃/Mo₂C-C, b) XPS survey spectra and elemental content for Mo₂C-C.

Comment 2: *Given the complicated nature of the current catalyst system containing multiple elements (C, N, O), we strongly suggest the authors employ a simplified unsupported system. Preparation of unsupported Mo₂N and Mo₂C through conventional ammonia/methane reduction method would enable more conclusive investigation of surface molybdenum species' catalytic behavior.*

Response to comment 2: We sincerely appreciate your valuable comments, which have greatly contributed to a deeper investigation into the catalytic behavior associated with surface reconstruction in molybdenum-based catalysts. We have synthesized unsupported Mo₂N and Mo₂C via ammonia and methane reduction methods. SEM and XRD characterizations confirm the successful synthesis of Mo₂N and Mo₂C (Supplementary Figures 29, 30). Raman spectroscopy was used to detect the oxide layers on the unsupported Mo₂N and Mo₂C surfaces, where stronger MoO₃ signals were observed on Mo₂N than on Mo₂C, which exhibits a consistent pattern with MoO₃/Mo₂N-C and MoO₃/MoO₂-C catalysts (Supplementary Figure 30). Additionally, CO₂ conversion tests before and after H₂ treatment show that H₂ treatment does not affect Mo₂C performance, but it enhances the conversion rate of Mo₂N, which also aligns with trends observed for MoO₃/Mo₂N-C and MoO₃/MoO₂-C catalysts (Supplementary Figure 31). This demonstrates that Mo₂N and Mo₂C exhibit identical surface reconstruction behavior to our MoO₃-supported catalysts in RWGS reactions. We appreciate your thoughtful input, which has significantly contributed to the development of our work. The corresponding details have been added in the revised manuscript and supplementary information, as shown below:

Page 14 in the revised manuscript: “Additionally, we synthesized unsupported Mo₂N and Mo₂C via ammonia and methane reduction methods (SEM, Supplementary Figure 29), with XRD characterization confirming perfect alignment of all lattice diffraction peaks with standard Mo₂N and Mo₂C reference patterns. Subsequent Raman analysis revealed oxide layers on the unsupported Mo₂N and Mo₂C surfaces, where Mo₂N exhibited stronger MoO₃ Raman signals than Mo₂C, which is consistent with trends observed for MoO₃/Mo₂N-C and MoO₃/MoO₂-C catalysts (Supplementary Figure 30). Furthermore, CO₂ conversion tests in RWGS reactions before and after H₂ treatment demonstrated unchanged performance for Mo₂C but enhanced conversion rates for Mo₂N, unequivocally establishing identical surface reconstruction behavior to our catalysts (Supplementary Figure 31).”

Supplementary Figure 29. SEM images of **a, b)** Mo₂N, **c, d)** Mo₂C prepared via ammonia and methane reduction.

Supplementary Figure 30. **a, b)** XRD patterns of Mo₂N and Mo₂C, **c)** Raman spectra of Mo₂N and Mo₂C prepared via ammonia and methane reduction.

Supplementary Figure 31. a, b) Catalytic results for CO yield under different temperatures under 60,000 mL·g_{cat}⁻¹·h⁻¹. (In a-b, n = 3 independent experiments, data are presented as mean values ± SD)

Comment 3: *Since the redox pathway likely involves the MoOC_x ↔ Mo₂C—a well-known process—the authors should expand their mechanistic discussion to include this aspect.*

Response to comment 3: We sincerely appreciate your valuable comments. For a deeper mechanistic discussion in considering the MoOC_x formation, we have synthesized the Mo₂C-C catalyst without MoO₃ coverage. The directly exposed Mo₂C makes it susceptible to oxidation by CO₂ or H₂O, forming MoOC_x. However, although our MoO₃/Mo₂C-C catalyst was deliberately passivated with O₂ during preparation, the fully oxidized MoO₃ surface remains highly resistant to forming MoOC_x species under RWGS reaction conditions. Furthermore, we conducted in situ mass spectrometry experiments using isotope labeling (¹³CO₂). The results revealed no carbon exchange mechanism on the MoO₃/Mo₂C-C catalyst; only ¹³CO was detected, and no ¹²CO was observed (Supplementary Fig. 36). This confirms that both the surface MoO₃ layer and the bulk Mo₂C phase in the MoO₃/Mo₂C-C catalyst remain highly stable during RWGS reaction, with no transformation into MoOC_x occurring. The corresponding data and discussions have been added in the revised manuscript and supplementary information, as shown below:

Page 17 in the revised manuscript: “Isotope labeling experiments have been employed to trace the origin of reactant components. We utilized a feed gas mixture of 2.5% ¹³CO₂ + 97.5% Ar and employed in situ mass spectrometry to distinguish the carbon source of the CO product. As shown in Supplementary Fig. 36, when ¹³CO₂ was pulsed over the MoO₃/Mo₂C-C catalyst in the absence of H₂, the C=O bond in ¹³CO₂ cleaved, generating ¹³CO; no ¹²CO was observed. This confirms that neither the surface MoO₃ layer nor the bulk Mo₂C in the MoO₃/Mo₂C-C catalyst underwent a carbon exchange mechanism during the RWGS reaction. Consequently, all carbon in the produced CO originates exclusively from CO₂.”

Supplementary Figure 36 Pulse experiments using isotope labelling in situ mass spectrometry with $^{13}\text{CO}_2$ for the $\text{MoO}_3/\text{Mo}_2\text{C-C}$ catalysts at 500 °C.

Comment 4: *The weak and ambiguous signals in situ Raman data (Fig.5) do not provide convincing evidence for the authors' conclusions.*

Response to comment 4: We appreciate your valuable comment. Indeed, the quality of the in situ Raman spectra was somewhat limited. Upon investigation, we found this issue was caused by wear of the quartz tube within the in situ Raman cell. As shown in Fig. 5a-c, after replacing the in situ Raman cell with a new one, we observed a significant increase in the intensity of all Raman signals. Furthermore, the conclusions drawn remain consistent with those previously obtained: surface reconstruction generates MoO_x on the $\text{MoO}_3/\text{Mo}_2\text{N-C}$ catalyst, MoO_2 on the $\text{MoO}_3/\text{MoO}_2\text{-C}$ catalyst, while MoO_3 on the $\text{MoO}_3/\text{Mo}_2\text{C-C}$ catalyst did not undergo reconstruction. Likely due to the improved signal quality, surface reconstruction forming MoO_2 on the $\text{MoO}_3/\text{MoO}_2\text{-C}$ catalyst was observed at 100 °C. Relevant details have been revised in the manuscript as follows:

Page 15-16 in the revised manuscript: “For the $\text{MoO}_3/\text{MoO}_2\text{-C}$ catalyst, the Raman spectra depicted in Fig. 5b indicate that the surface MoO_3 structure has already been reduced to MoO_2 at 100°C. The characteristic peaks attributed to MoO_2 diminish with further temperature increase to 500 °C, suggesting that the surface MoO_2 is likely to constitute the principal active phase in the $\text{MoO}_3/\text{MoO}_2\text{-C}$ catalyst.”

Revised Fig. 5a-c) In-situ Raman spectroscopy results of MoO₃/Mo₂N-C, MoO₃/MoO₂-C, and MoO₃/Mo₂C-C under the RWGS reaction.

Comment 5: *The manuscript contains some minor writing errors that require correction, particularly in Page 8 (lines 144-145) and Figures 4d and 4e.*

Response to comment 5: We thank the reviewer for this kind reminder. We sincerely apologize for the oversight of the writing errors in our manuscript. We have thoroughly reviewed the entire manuscript and Supporting Information to ensure no related issues remain.

Reviewer #3 (Remarks to the Author):

“Molybdenum-based catalysts are investigated for CO₂ hydrogenation. In particular, the RWGS is the thermal catalytic process considered to explore. The analysis of the surface reconstruction and its consequences in catalytic performance. Analyzing the surface reconstruction is very important because it may conditionate the performance of the systems with the time, even promote the passivation. The authors emphasize that this study constitutes the first time the formation of MoO_x sites is reported and this could pave a way to design heterogeneous catalysts by constructing highly active metal oxides and adjusting the reducibility of substrates.”

Comment 1: *Mo₂N and Mo₂C are the substrates that (if understood correctly) form coherent 2D nanosheet nanostructures. What is the thickness of these systems? One may wonder about these are*

3D carbides or MXenes with a certain thickness. The structural analysis shows Mo_2N (220), MoO_2 (-211), and Mo_2C (110), but the schematic illustration (Fig 1 a) shows the Mo_2X (111) ($X=\text{C}, \text{N}$) passivated by MoO_x .

Response to comment 1: We appreciate your valuable comment, which is crucial for enhancing the quality of our work. The morphologies of our $\text{MoO}_3/\text{Mo}_2\text{N-C}$, $\text{MoO}_3/\text{MoO}_2\text{-C}$, and $\text{MoO}_3/\text{Mo}_2\text{C-C}$ catalysts exhibit large nanosheets with notably high layer thicknesses. Measurements reveal thicknesses of 395 nm, 430 nm, and 420 nm, respectively (Supplementary Figure 1). On these nanosheet structures, the surface oxidation layer could be within several nanometers, as indicated by the TEM result. This structure is significantly different from MXene materials.

It should be noted that our DFT computational models were constructed using the (1 1 1) crystal plane of Mo_2N . This plane was selected because the (1 1 1) facet is the most abundant surface in molybdenum nitride (Mo_2N), and low-index planes exhibit greater stability (*J. Phys. Chem. C* **2018**, 122, 21039-21046). While the (2 2 0) plane appears more distinct in HRTEM images due to imaging angles, this does not imply its predominance. To minimize the influence of variables (e.g., crystal system differences), we deliberately selected the most comparable crystallographic planes for modeling. This approach allows us to effectively highlight the differences in surface reconstruction among the three catalysts.

Supplementary Figure 1. SEM images of a) $\text{MoO}_3/\text{Mo}_2\text{N-C}$, b) $\text{MoO}_3/\text{MoO}_2\text{-C}$, c) $\text{MoO}_3/\text{Mo}_2\text{C-C}$ structures.

Comment 2: The connection between experiments and theory is fantastic. The computational strategy followed is reasonable and well-considered. It is important to see that redox mechanism is the most

*favorable respect to other routes. This aspect has been also observed in MXenes in studies published by Morales-García et al. It is important to see that the authors explore the potential energy landscape of the reaction including thermodynamic and kinetics aspects. They are now in a privilege situation of move and step further performing **microkinetic simulations to get the reaction rates and TOFs values and compare directly with their experiments (Fig, 4 g, h, and i). I strongly recommend to add this multiscaling analysis and I have curiosity the matching with experiments***

Response to comment 2: Thank you for this valuable suggestion. We strongly agree that incorporating microkinetic modeling would significantly enhance the matching between experiments and theoretical results. However, we have to apologize that neither our team nor our collaborating group possesses the necessary expertise to perform such calculations. We are currently exploring collaborations with international research groups specializing in microkinetic modeling while simultaneously learning the methodology ourselves. However, we are afraid that obtaining reliable microkinetic simulation data may require a long time for us. Although we cannot include microkinetic simulations at present, some of our current DFT calculations and characterization data could confirm most of our main claims: We evaluated the relative capabilities for CO₂ adsorption (Fig. 3a-c) and CO₂ activation (Fig. 3d-g) across three catalysts, which indicate that MoO₃/Mo₂N-C possesses a significantly stronger activation ability for CO₂ than the other catalysts, aligning well with the trends observed experimentally (CO₂ reaction orders and CO₂ conversion Fig. 4d, i). Our DFT predictions consistently match the experimental findings, indicating that microkinetic modeling would serve as a complementary enhancement rather than a determinant of our core conclusions. We sincerely hope that you can understand the current difficulties in doing microkinetic simulations, and we hope that we can include such simulations in our future related research.

Point-by-point response to the comments of the reviewers for the publication “Unraveling surface sensitivity for generating metastable active site in molybdenum-based catalysts for CO₂ hydrogenation”, manuscript ID: NCOMMS-25-25695A.

REVIEWER COMMENTS

Reviewer #2 (Remarks to the Author):

Comment 1: *The authors emphasized that activity analysis indicates the formation of metastable MoO_x species on MoO₃/Mo₂N-C, which contributes to its unprecedented RWGS performance. However, XPS results show that Mo₂N species remains the dominant component on the surface. Thus, the respective catalytic roles of MoO_x and Mo₂N remain unclear. I strongly recommend that the authors further compare the RWGS activity over the passivated MoO₃/Mo₂N-C catalyst, the H₂-pretreated MoO₃/Mo₂N-C catalyst, and the fresh (non-passivated) MoO₃/Mo₂N-C sample to clarify the true catalytic nature of the Mo₂N catalyst.*

Response to comment 1: We appreciate your helpful suggestion for enhancing the quality of our manuscript. Based on your suggestions, we conducted additional experiments to comprehensively compare the RWGS activity, reaction orders, and activation energies of the H₂-pretreated MoO₃/Mo₂N-C catalyst (MoO₃/Mo₂N-C), the passivated MoO₃/Mo₂N-C catalyst (denoted as passivated MoO₃/Mo₂N-C), and the fresh MoO₃/Mo₂N-C sample (denoted as Mo₂N-C) under identical conditions. This comparative analysis provides a holistic understanding of the respective catalytic mechanisms of MoO_x and Mo₂N, elucidating the intrinsic catalytic nature of molybdenum nitride catalysts in the RWGS reaction.

As shown in **Supplementary Fig. 29a and b**, the Mo₂N-C catalyst, due to the absence of an oxide layer, exhibits a significant decrease in both CO₂ conversion and CO formation rate compared to the MoO₃/Mo₂N-C catalyst. Meanwhile, the passivated MoO₃/Mo₂N-C catalyst, which contains an oxide layer but has not undergone H₂ pretreatment, shows very low MoO_x content in its oxide layer, resulting in lower CO₂ conversion and CO formation rate than the MoO₃/Mo₂N-C catalyst. These results demonstrate that the decisive factor for the RWGS activity of the MoO₃/Mo₂N-C catalyst is the MoO_x species generated after H₂ treatment, rather than MoO₃. By comparing the H₂ reaction orders of the three catalysts, it is evident that the Mo₂N-C catalyst (which lacks an oxide coating and fully exposes

Mo₂N sites) has the lowest H₂ reaction order and the strongest ability to activate H₂. This strongly confirms that Mo₂N serves as the active site for H₂ activation in the RWGS reaction (Supplementary Fig. 29c). Analysis of the CO₂ reaction orders of the three catalysts reveals that the Mo₂N-C catalyst, without an oxide layer, has the highest CO₂ reaction order, indicating its weakest ability to activate CO₂. In contrast, the MoO₃/Mo₂N-C catalyst, possessing MoO_x species, exhibits the strongest CO₂ activation capability (Supplementary Fig. 29d). Supplementary Fig. 29d clearly demonstrates that the surface oxide layer is responsible for CO₂ activation, and MoO_x significantly enhances this ability. As shown in Supplementary Fig. 29e, the MoO₃/Mo₂N-C catalyst has the lowest activation energy, while the passivated MoO₃/Mo₂N-C catalyst shows the highest activation energy. This suggests that CO₂ activation by the MoO₃ oxide layer may be more dependent on high temperature, whereas MoO_x can markedly reduce the reaction's dependence on temperature. Supplementary Fig. 29f provides a concise and clear overview of the comparison of RWGS activity, reaction orders, and activation energies among the three catalysts. The corresponding details have been added in the revised manuscript, shown below:

Page 14 in the revised manuscript: “To elucidate the catalytic nature of MoO₃/Mo₂N-C, we comprehensively compared the RWGS performance metrics of the MoO₃/Mo₂N-C catalyst, the passivated MoO₃/Mo₂N-C catalyst (lacking MoO_x), and the Mo₂N-C catalyst (lacking the surface MoO₃ oxide layer). As shown in Supplementary Fig. 29a-e, the MoO₃/Mo₂N-C catalyst exhibits the highest CO₂ conversion and CO formation rate, lowest CO₂ reaction order, and activation energy. These results suggest that MoO_x, rather than MoO₃, serves as the primary active species in the catalytic system. In contrast, the Mo₂N-C catalyst demonstrates the lowest H₂ reaction order, indicating the strongest H₂ activation capability when Mo₂N is fully exposed. The CO₂ and H₂ reaction orders, as well as the activation energy, reflect the dependence of the catalytic reaction on CO₂ concentration, H₂ concentration, and temperature, respectively. Supplementary Fig. 29f visually summarizes the roles of Mo₂N and MoO_x in the RWGS reaction mechanism, demonstrating that the exposure of Mo₂N reduces the reaction's dependence on H₂ concentration, while MoO_x significantly reduces its dependence on both CO₂ concentration and temperature.”

Supplementary Figure 29 a) CO₂ conversion at various temperatures, b) Catalytic reaction rates at various temperatures. c) Kinetic orders of H₂ for MoO₃/Mo₂N-C, passivated MoO₃/Mo₂N-C and Mo₂N-C catalysts, d) Kinetic orders of CO₂ for MoO₃/Mo₂N-C, passivated MoO₃/Mo₂N-C and Mo₂N-C catalysts. e) E_a for MoO₃/Mo₂N-C, passivated MoO₃/Mo₂N-C and Mo₂N-C catalysts. f) E_a for MoO₃/MoO₂-C and MoO₂-C catalysts. Performance comparison on the CO₂ conversion, reaction rates, Kinetic orders of H₂ and CO₂, and E_a of MoO₃/Mo₂N-C, passivated MoO₃/Mo₂N-C and Mo₂N-C catalysts. (In a-b, n = 3 independent experiments, data are presented as mean values ± SD)